



# New Particle Formation in the South Aegean Sea during the Etesians: importance for CCN production and cloud droplet number

Kalkavouras P.[1], Bossioli E.[1], Bezantakos S.[2], Bougiatioti A.[3,7], Kalivitis N.[3], Stavroulas I.[3], Kouvarakis G.[3], Protonotariou A. P.[1], Dandou A.[1], Biskos G.[4,5], Mihalopoulos N.[3,5,6], Nenes A.[6,7,8,9], Tombrou M.[1*]

[1] Department of Physics, Univ. of Athens, 15784 Athens, Greece
[2] Dept. of Environment, Univ. of the Aegean, Mytilene 81100, Greece
[3] Env. Chemical Processes Lab., Dept. of Chemistry, Univ. of Crete, Heraklion 71003, Greece
[4] Fac. of Civil Engineering and Geosciences, Delft Univ. of Technology, Delft 2628 CN, The Netherlands
[5] Energy Environment and Water Research Center, The Cyprus Institute, Nicosia 2121, Cyprus
[6] Inst. of Env. Research & Sustainable Development, Nat. Observatory of Athens, Greece
[7] School of Earth and Atmospheric Sciences, Georgia Institute of Technology, Atlanta, GA, USA
[8] School of Chemical and Biomolecular Engineering, Georgia Institute of Technology, Atlanta, GA, USA
[9] Institute for Chemical Engineering Science, Foundation for Research and Technology Hellas, Patra, Greece

*Correspondence to*: Maria Tombrou (mtombrou@phys.uoa.gr)

**Abstract.** We examine the concentration levels and size distribution of submicron aerosol particles along with the concentration of trace gases and meteorological variables over the central (Santorini) and south Aegean Sea (Crete) from 15 to 28 July 2013, a period that includes Etesian events and moderate northern winds. Particle nucleation bursts were recorded during the Etesian flow at both stations, with those observed at Santorini reaching up to $1.5 \times 10^4$ particles cm$^{-3}$. On Crete (at Finokalia station), the fraction of nucleation-mode particles was diminished, but a higher number of Aitken-mode was observed as a result of the downward mixing and photochemistry. Aerosol and photochemical pollutants covaried throughout the measurement period: lower concentrations were observed during the period of strong Etesian flow (e.g. 43 - 70 ppbv for ozone, 1.5 -5.7 µg m$^{-3}$ for sulfate), but were substantially enhanced during the period of moderate winds (i.e., increase of up to 32% for ozone, and 140% for sulfate). To understand how new particle formation (NPF) affects cloud formation, we quantify its impact on the CCN levels and cloud droplet number concentration. We find that NPF can double CCN number (at 0.1% supersaturation) but the resulting strong competition for water vapor in cloudy updrafts decreases maximum supersaturation by 14% and augments the potential droplet number only by 12%. Therefore, although NPF events may strongly elevate CCN numbers, the relative impacts on cloud droplet number (compared to pre-event levels) is eventually limited by water vapor availability and depends on the prevailing cloud formation dynamics and the aerosol levels associated with the background in the region.



**Keywords.** Ozone concentrations, Aerosol particle concentrations, Particle size distributions, Chemical composition, New particle formation, CCN production, droplet number, WRF-Chem simulations

# 1 Introduction

During summer and early autumn (warm period), the circulation over the Eastern Mediterranean (EM) is dominated by a persistent northerly flow known as Etesians (Tyrlis et al., 2013). Under the prevalence of the Etesians, the advection of the air masses is pronounced over the EM rendering the atmospheric conditions as the most important factor for high concentrations of gases and aerosol particles even in remote areas. The scientific interest over the Aegean Sea (AS), which is part of the EM, has led to a number of experimental campaigns, during the warm period (Mihalopoulos et al., 1997; Paronis et al., 1998; Formenti et al., 2002a, b; Kouvarakis et al., 2002; Lelieveld et al., 2002; Zerefos et al., 2002), focusing initially on the interpretation of ozone ($O_3$) enhancement under the Etesian regime. Apart from the simultaneous contribution of local and distant sources in the area, in the presence of enhanced photochemistry, strong subsidence was also identified in most of these events (Kallos et al., 1998, 2007; Lelieveld et al., 2002; Salisbury et al., 2003; Kalabokas et al., 2007, 2008, 2013; Kanakidou et al., 2011; Bossioli et al., 2016). Airborne measurements performed during an Etesian outbreak (Aegean-GAME campaign; Tombrou et al., 2015) have clearly shown that neutral to stable atmospheric conditions prevailed over the north and central AS, with reduced friction velocities and absolute turbulent fluxes (momentum or heat) cumulating the concentrations below the planetary boundary layer (PBL) and mainly inside the shallow Marine Atmospheric Boundary Layer (MABL). Unstable conditions were found only over southeast AS, in the vicinity of Crete, resulting in enhanced friction velocities and large positive values of sensible heat flux.

Long-term aerosol observational studies in the EM have been mainly performed based on ground measurements collected at Finokalia, a remote coastal site in the northeastern part on the island of Crete (Bardouki et al., 2003; Eleftheriadis et al., 2006; Lazaridis et al., 2006; Gerasopoulos et al., 2007; Kalivitis et al., 2008, 2014, 2015; Koulouri et al., 2008; Querol et al., 2009; Pikridas et al., 2010, 2012) with a few more at Akrotiri station on western Crete (Lazaridis et al., 2008; Kopanakis et al., 2012, 2013). Most of these ground-based observations indicate that the mass of fine aerosols presents a summer maximum, without any particular seasonal preference about their occurrence. These fine aerosols have been related to regional sources of pollution enhanced by long-range transport (LRT) during the Etesian flow. In particular, a mixture of anthropogenic (Koçak et al., 2011), biogenic (Im and Kanakidou, 2012) and biomass burning emissions (Sciare et al., 2008; Bougiatioti et al., 2014) originating mainly from the Balkans and the central and Eastern Europe, result in enhanced aerosol concentrations in the southern AS (Kalivitis et al., 2014).

Short-lived events of small number young Aitken particles (18–50 nm) were first recorded at Finokalia by Kalivitis et al. (2008), arriving with low speed from the west, during autumn. Thereafter, new particle formation (NPF) events have





been frequently observed at Finokalia (Manninen et al., 2010; Ždímal et al., 2011; Pikridas et al., 2012; Kalivitis et al., 2014, 2015) and Akrotiri (Kopanakis et al., 2013) stations during different periods of the year, but more frequently during winter than summer. According to the literature, the NPF events are favored when airmasses are enriched by a reactant (probably $NH_3$), prior reaching the site of Finokalia (Pikridas et al., 2012; Kalivitis et al., 2014). During the Etesians in particular, the particle size distributions were centered on the lower end of the accumulation-mode size range (Kalivitis et al., 2014). This was partly attributed to the fast oxidation of $SO_2$ that resulted mainly in acidic particles and to the increase of the condensation sink, which is the main limiting factor suppressing the events during the summer (Pikridas et al., 2012). It has been only recently shown that NPF events could occur at Finokalia during Etesians (Kalivitis et al., 2015). A large number of $PM_1$ particles (of the order of $10^4$ $cm^{-3}$) were also observed at the northeastern AS during an Etesian outbreak (Tombrou et al., 2015), whereas high number concentrations of nucleation-mode particles observed in the north AS by Triantafyllou et al. (2016), have been associated with polluted air masses transported from Istanbul.

A natural question therefore, is to understand the track record of the air masses passing over the Aegean before arriving at Finokalia. In particular, we need to elucidate the atmospheric and chemical processes that affect ageing of the air masses passing over the AS maritime area between the Cyclades and Crete, and furthermore, examine whether NPF events observed at Finokalia would be stronger over the central Aegean during the northern Etesian flow. What is more, evidence that the NPF events are associated with an increase in the concentration of cloud condensation nuclei (CCN) production in the EM atmosphere, has been recently demonstrated based on simultaneous measurement of particle number size distributions, CCN properties and aerosol chemical composition (Kalivitis et al., 2015). Bougatioti et al. (2009, 2011) have shown high CCN concentrations at Finokalia, from air masses coming from the Balkans, during a period representative of an Etesian regime. However, no study to date has actually focused on understanding the increase in cloud droplet number that results from NPF, which is the true microphysical link between NPF and the aerosol indirect effect.

Driven by the above arguments, we chose to perform measurements at a remote site on Santorini, which is located within the same path of airmasses reaching the station of Finokalia, during the Etesians. Our aim is to elucidate both atmospheric and chemical processes that affect ageing of the air masses passing over the AS before reaching its southern edge, the island of Crete. Continuous ground measurements of particle properties, concentration of gaseous species, and meteorological variables were simultaneously collected on Santorini and Crete. During this short-term campaign (15-28 July 2013) intense bursts of nucleation-mode particles were observed at both sites. The synoptic wind flow and boundary layer dynamics as well as the atmospheric chemical composition that favor the enhanced NPF events during the Etesian flow are examined in this study. To understand how NPF could affect cloud formation throughout its evolution, we quantify its impact on CCN levels, cloud droplet number concentration (CDCN) and supersaturation formed in clouds that develop before, during and after NPF events at both ground sites. Complementary to this analysis, wind patterns and atmospheric chemical composition based on WRF-Chem mesoscale model simulations, are presented.



## 2 Methodology

### 2.1 Experimental Observations

Ground level measurements were conducted simultaneously at two remote coastal areas (cf. Fig. 1), from 15 to 28 July 2013: on the island of Santorini (at Ag. Artemios; 36° 26' N, 25° 26' E) and at the monitoring station of Finokalia on the island of

Crete (35° 20' N, 25° 40' E; http://finokalia.chemistry.uoc.gr; Mihalopoulos et al., 1997). Ag. Artemios (hereafter referred to as Santorini) is located at an elevation of 153 m above sea level (asl), while Finokalia on the top of a hill at 260 m asl. Both measuring sites are far from any large city or anthropogenic activity, and are close to the sea; Finokalia is facing the sea within a sector of 270º to 90º, whereas the station on Santorini within a sector of 340º to 120º.

The Finokalia monitoring station houses a suite of instruments for measuring the meteorological parameters, the

concentrations of gaseous species, as well as the physical properties and chemical composition of atmospheric particles. We used an $O_3$ analyzer (Thermo Electron model 49I), a Scanning Mobility Particle Sizer (SMPS; TROPOS Type, Wiedensohler et al., 2012) with a TSI-3772 condensation particle counter (CPC) for measuring the size distribution of aerosol particles having diameters from 9 to 848 nm (scanned range), and an Aerodyne Research Inc. Aerosol Chemical Speciation monitor (ACSM; Ng et al., 2011), for measuring the mass and chemical composition ( $SO_4^{2-}$, $NO_3^-$, $NH_4^+$, Cl⁻, and organics) of non-

refractory submicron aerosol particles. A TSI SMPS (Model 3034) measured the size distribution of particles having diameters from 10 to ca. 500 nm at Santorini. The concentrations of gaseous species were also measured using standard instrumentation, including an $O_3$ analyzer (Photometric M400E) and a dual channel chemiluminescence analyzer for nitrogen oxides (NO, $NO_2$; Photometric M200E) and a fluorescence analyzer for sulfur dioxide ($SO_2$; Photometric M100E). An overview of the instruments used for the measurements is provided in Table 1.

### 2.2 Regional modeling

The WRF-Chem version-3.3 mesoscale model (Grell et al., 2005) is used to understand the dominant meteorological regimes and the regional characteristics of the aerosol during the sampling period. Simulations were performed by applying triple nesting: the outmost 1$^{st}$ domain covers the extended area of Europe (spatial resolution 0.5º × 0.5º); the first level of nesting 2$^{nd}$ nested domain covers the extended area of Greece and Italy (0.167º × 0.167º) and the innermost nesting is centered on

the extended area of Greece (0.056º × 0.056º). The RADM2 chemical mechanism is used to simulate the gas phase chemistry (Stockwell et al., 1990), while the Modal Aerosol Dynamics Model for Europe (MADE) (Ackermann et al., 1998) and Secondary Organic Aerosol Model (SORGAM) (Schell et al., 2001) are used to simulate inorganic and secondary organic aerosol (SOA), respectively. The size distribution is described by three log-normal modes (Aitken, accumulation, and coarse). New particle formation is treated with the Kulmala et al. (1998) parameterization of sulfuric acid nucleation. New

particles are assigned to the Aitken mode with a diameter of 3.5 nm. Low-vapor-pressure gas-phase species condense onto existing particles at a rate determined by Binkowski and Shankar (1995) method. The aerosol species treated by the modules



are the main inorganic ions (SO$_4^{2-}$, NO$_3^-$, NH$_4^+$, Na$^+$, Cl$^-$), elemental carbon (EC), primary organic aerosols (POA), SOA, a primary unspeciated PM$_{2.5}$ fraction covering all the unspeciated/unknown fine particles (PM$_{2.5\text{-unsp}}$), and three species for the coarse mode (i.e., anthropogenic, marine, and soil derived aerosols). For the fine particles fraction, each model species has an Aitken-mode and an accumulation-mode component. For anthropogenic emissions from Europe (1$^{st}$ and 2$^{nd}$ domains) we use the EMEP database while for Greece (3$^{rd}$ domain) we employ the national emission inventory (Tombrou et al., 2009). Natural (biogenic and sea-salt) emissions are calculated on-line within the WRF-Chem model. Simulations were performed from 12 to 29 July. An extended evaluation of WRF-Chem model against airborne and ground observations over the AS during the Etesians is presented in Bossioli et al. (2016). Under long-range transport conditions, the model successfully simulates CO, O$_3$, sulfate, and ammonium concentrations while it underestimates the aerosol carbonaceous fraction, which is mostly organic matter.

Back trajectories determined using the HYSPLIT4 Model (Hybrid Single-Particle Lagrangian Integrated Trajectory), developed by the Air Resources Laboratory of the National Oceanic and Atmospheric Administration (NOAA) (Draxler and Rolph, 2015) to reveal the origin of air masses reaching the study area. The back-trajectories, initialized with meteorological conditions from GDAS (0.5$^{o}$), were computed at several heights. All three-dimensional trajectories were computed with an end point either at Santorini or Finokalia station.

## 3 Results and Discussion

### 3.1 Prevailing atmospheric and air quality conditions

Northern winds prevailed over the AS throughout the entire campaign. Based on the simulated wind patterns at 100 m above ground level (agl) throughout Greece (cf. Fig. S1 in the supplementary material) and on the sea level pressure fields (NCEP/NCAR; Fig. S2) the 17 - 18 July and 22 - 24 July are periods of strong Etesian winds (Brody and Nestor, 1985; Kotroni et al., 2001; Anagnostopoulou et al., 2014). Hereafter, we refer only to the second period, as higher aerosol number concentrations were measured at both stations, but also because there were no O$_3$ measurements at Santorini, during the first period. Immediately after the second period, another characteristic period followed (25 - 27 July), having a similar pressure pattern with the previous two; the pressure gradient over the Dardanelles was weaker. Back trajectory analysis of the air masses sampled at both stations indicates almost the same source regions, for both periods (Figs. 2, S3). However, different conditions prevailed during these two periods altering mainly the last part of the journey of the airmasses, over the AS. From 22 to 24 July, strong northern wind speeds prevailed (> 10 m s$^{-1}$) with the wind direction forming the characteristic 'ring-shape' (Fig. S1) of the Etesian flow around Turkey (Tyrlis and Lelieveld, 2013). From 25 to 27 July, moderate wind speeds (up to ca. 8 m s$^{-1}$) with northeasterly surface flow displayed over the central and southern AS, while stagnant conditions prevailed at the north (Fig. S1).

The measured wind speeds at Finokalia station exceeded 9 m s$^{-1}$, and the wind direction was mainly from west - southwestern during the daytime hours (Fig. 3) owing to topographic features that steer the prevailing direction towards the



west/southwest direction. Capturing this local feature is a known challenge for regional models (e.g., Gauss et al., 2011; Im et al., 2011; Hodneborg et al., 2012). At the same time, the simulated wind direction at Santorini station exhibited a northern direction, with wind speeds exceeding 8 m s$^{-1}$ during the daytime hours (Fig. 3).

The number concentrations for the three particle modes (nucleation, Aitken and accumulation) together with the O$_3$ concentrations are shown for both periods at Santorini and Finokalia stations in Fig. 4. Simultaneous routine meteorological measurements, such as surface temperature and relative humidity, are also provided for each station. Apart from the region-wide differences, intense bursts in the concentration of nucleation-mode particles having diameters smaller than 25 nm were observed at both stations during the period of Etesians; it should be noted that these events were not observed at any of the stations during the period of moderate winds (Fig. 4). In the subsequent sections the different characteristics and processes prevailing under strong or moderate northerly wind flow are explored aiming to elucidate the interconnection between the two stations.

## 3.2 Ozone concentrations

During the Etesian flow period, O$_3$ levels at Santorini and Finokalia stations ranged between 38 and 66 ppbv and 43 to 70 ppbv, respectively (Figs. 4, S4, Table 2); these levels are consistent with previous measurements (57 ± 4 ppbv) inside the MABL for Etesian flow carried out during the Aegean-GAME campaign. The values also agree with the climatological values recorded over the greater area during summer (Gerasopoulos et al., 2005; Kalabokas et al., 2007, 2013). The less pronounced diurnal cycle at Finokalia station, compared to Santorini (Fig. S4, is attributed to a shallower and more stable MABL over Santorini compared to Crete during Etesian flow conditions (Tombrou et al., 2015) that favors higher primary concentrations and thus O$_3$ scavenging at Santorini, especially when the MABL collapses during nighttime. In the vicinity of Crete, the MABL becomes less stable, due to larger sea surface temperatures (SST) existing southern of Santorini. This fact, together with the topography (i.e., Crete forms a mass of land that is located perpendicular to the Etesian flow), enhances the mixing and the downward transport from the above rich in O$_3$ concentrations layer. During the moderate winds, high O$_3$ levels (the highest concentrations of the summer in 2013) were measured at both stations, ranging between 50 and 99 ppbv (Figs. 4, S4, Table 2). At both stations the highest values were observed on 26 July. The lower winds over the northern AS contributed to O$_3$ accumulation at this area, explaining the high O$_3$ concentrations at both stations. The maximum O$_3$ concentration observed (but simulated as well) at Finokalia had a 4-h delay compared to that observed at Santorini. Simulations confirm that the air masses received at both stations during the strong wind period are of the same origin, and representative of Etesian flow conditions (Fig. S3) albeit with a small O$_3$ under-prediction (average bias during afternoon hours up to -21% on 23 and -15% on 24 July, Fig. 5) while they also indicate an O$_3$ increase during the moderate winds, especially in the southern AS, but also underpredicted (average bias during afternoon hours up to -24% on 26 July, Fig. 5). The model underestimation mainly arises from inaccuracies of the emissions inventory as well as to fixed chemical boundary conditions, representative of clean environment conditions (McKeen et al., 1991). Bossioli et al. (2016) have shown that when realistic representation of the stratosphere-troposphere exchange processes are implemented in WRF-Chem



simulations through the time-varying chemical boundary conditions from the MOZART global chemical transport model, this resulted to a significant $O_3$ increase inside the PBL (up to 40%), during Etesians.

### 3.3 Aerosol mass and number concentrations

Figure 6 shows the non-refractive submicron aerosol concentrations measured at Finokalia during the whole experimental period. In general the inorganic and organic mass concentrations had a similar behavior with $O_3$ during most of the experimental period (Fig. 4). During Etesian conditions, the $PM_1$ mass concentrations were reduced roughly by a factor of two compared to those during the moderate wind period (Table 2), and were in the range of concentrations measured in the framework of the Aegean-GAME campaign. However, despite that the concentrations of all four species ($SO_4^{2-}$, $NO_3^-$, $NH_4^+$ and organics), were substantially decreased during 23-24 July, the organic fraction exhibited a relative increase, especially at the beginning of this period. Due to lack of data at Santorini, simulated $PM_{2.5}$ mass concentrations are used for the analysis. The modeled concentrations for sulfate are about 3 μg m$^{-3}$, at both stations at 09:00 LST (Fig. 7) quite close to the measured values at Finokalia (Fig. 6). Similar to the case of $O_3$, the two stations are located along the less polluted airflow over the AS, on the boundary of the heavily polluted air masses over the eastern AS (Fig. 7) and downward of a thin plume, that starts over the northwestern Asian Turkish coast.

During the moderate wind period (Fig. 3), the aerosol mass concentrations at Finokalia were substantially higher (Table 2; Fig. 6). The increased concentrations were retained until noon of 27 July for sulfate and ammonium, while those of organics continued to increase further until the end of the campaign. The modeled spatial distribution of sulfate concentrations was nearly uniform over the AS, while as for ozone their concentrations increased offshore of the northeastern coast of Crete due to the ageing of air masses in combination with the strong impact of the topography (Fig. 7). The simulated mass concentrations of secondary inorganic fine aerosols are in agreement with the measured values at Finokalia station.

In contrast to the fine aerosol mass concentrations, their total number concentrations were substantially increased, reaching continental levels during Etesian flow conditions (from 1.5 x 10$^3$ to 1.5 x 10$^4$ cm$^{-3}$ at Santorini and from 2.4 x 10$^3$ to 7.5 10$^3$ cm$^{-3}$ at Finokalia; Table 2, Fig. S5). The Aitken-mode particles followed a similar diurnal variation at both stations, ranging from $4.4 \times 10^2$ to $7.7 \times 10^3$ cm$^{-3}$ and peaking around noon. Accumulation-mode particles were higher at Finokalia. The total particle number concentration measured within the MABL of eastern AS during Aegean-GAME campaign under similar atmospheric conditions were on average $8 \times 10^3$ cm$^{-3}$, with almost 20% ($1.4 \pm 1.2 \times 10^3$ cm$^{-3}$) being in the 20-50 nm size range (Tombrou et al., 2015). Greater differences were observed for the nucleation-mode particles (i.e. particles having diameters smaller than 25 nm), with sudden concentration bursts observed at both stations (Fig. 4). On 23 July, a nucleation-mode burst was recorded, reaching number concentrations up to $1.3 \times 10^4$ cm$^{-3}$ at Santorini and almost $1.4 \times 10^3$ cm$^{-3}$ at Finokalia. A second event, but of lower intensity, was recorded on 24 July. It is worth mentioning that apart from the strong winds and lower temperatures, this period is considered humid (relative humidity values reaching up to 80% at Finokalia



station) in comparison to the period of moderate winds (Fig. 4). The nucleation-mode particles shift gradually towards larger sizes in a banana-shape pattern at both stations, as shown in Fig. 8. The number of particles remained high for several hours at Santorini (cf. Fig. 8), indicating regional NPF (Kulmala et al., 2012).

The associated growth rates (*GR*) for particles that increased in size from 10 to 25 nm were estimated to be 3.06 nm h$^{-1}$ at Santorini and 2.05 nm h$^{-1}$ at Finokalia on 23 July, and 2.08 nm h$^{-1}$ and 1.76 nm h$^{-1}$, respectively, on 24 July. The average *GR* for particles increasing in size from 7 to 20 nm at Finokalia was reported to be substantially higher (7.5 ± 5.8 nm h$^{-1}$) by Pikridas et al. (2012), with the highest daily *GRs* observed during the hottest months of the year (May to July 2008). It should be mentioned, however, that the nucleation events reported in that study were mainly related to air masses spending most of the time over the island of Crete, which is not the case for the observations reported here. Neglecting any coagulation losses, formation rates, $J_D$, can be computed as $J_D = \Delta N \, \Delta t^{-1}$, where $\Delta N$ is the number increase of nucleated particles (for a defined size range) during the NPF event. For the two consequent events at Santorini, $J_D$ for particles having diameters from 10 to 25 nm ranged between 2.12 cm$^{-3}$ sec$^{-1}$ (23 July) and 1.16 cm$^{-3}$ sec$^{-1}$ (24 July; Fig. 8). At the station of Finokalia, $J_D$ was lower for particles between 9and 25 nm, ranging between 0.27 (23 July) and 1.01 cm$^{-3}$ sec$^{-1}$ (24 July; Fig. 8). The similarity between the $J_D$ rates at the two sites on 24 July indicate a region-wide NPF event has occurred, yet the rates taken a day earlier are markedly different and thus, indicating a local event. However, we will show later (section 3.4) that this is not the case and more information needs to be taken into account.

Under moderate winds, the total fine aerosol number concentrations were considerably lower than those during the Etesians (from $1.4 \times 10^3$ to $2.9 \times 10^3$ cm$^{-3}$ at Santorini and from $2.6 \times 10^3$ to $5.1 \times 10^3$ cm$^{-3}$ at Finokalia; Fig. S5). Particles in the nucleation mode were absent, while the concentrations in the Aitken mode were substantially lower at both stations, varying from $3.2 \times 10^2$ to $4.1 \times 10^3$ cm$^{-3}$ (Fig. S5). The particle concentrations in the accumulation mode at Santorini had a comparable variation to that of the Aitken-mode , while they were apparently always higher at Finokalia.

### 3.4 Spatial extent of NPF event

The synoptic wind flow and boundary layer dynamics as well as the chemical atmospheric background conditions that favor the enhanced NPF events during the Etesian flow are further examined here. This type of event could be characterized as "type A" according to Boy and Kulmala (2002), owing to the sudden appearance of nucleation-mode particles and their consistent growth for at least 1 hour. The horizontal scale of this event was estimated based on air mass back-trajectory analysis (Hussein et al., 2009), taking into account the time during which measurements at the site indicate a distinct nucleation mode. Following Birmili et al. (2003), HYSPLIT4 back-trajectory calculations started at the time when a nucleation mode was first distinguishable from the Aitken mode at Santorini and were performed for each subsequent hour until the two modes merged (nucleation duration). Following Crippa and Pryor (2013), the duration of NPF was based on the geometric mean diameter of particles with sizes between 10 and 100 nm and from 30 to 100 nm; an event is said to initiate when the difference between the two geometric mean diameters becomes maximum and ends when this difference is less



than 15% (Fig. S6). Assuming a linear $GR$ (Lehtinen and Kulmala, 2003), this approach showed that the ca. 10-nm particles (the smallest particles we could detect with our instrumentation) were able to grow up to 60 nm within 4.5 h of initial detection. This $GR$ was then used to calculate the minimum spatial scale. On 23 July, the distance covered by the back-trajectories within 4.5 h (starting when the nucleation-mode burst was first recorded at Santorini) spans at least over 250 km

to the northeast of Santorini in the center of AS, upwind of the Cyclades complex (at 39° N; cf. Fig. 2). A couple of hours before the sunrise these back-trajectories (both below and above 500 m agl) are observed over the northwestern Asian forest peninsula of Turkey (Fig. 2; area marked as A in Fig. 1), having previously passed (at higher altitudes ≥ 1000 m agl) from the greater area of Istanbul and the west coast of the Black Sea. WRF-Chem simulations provide additional evidence of this flow pattern on 23 July and show that the prevailing strong wind favors the LRT of air masses, including those originating

from higher levels. A similar spatial extent also occurs during the less intense event on 24 July, although this starts with two hours delay (Fig. 2). Air masses are better mixed throughout the boundary layer covering a broader area over Asian Turkey on 24 July.

The plume along the back-trajectories characterized by high concentrations of ultra-fine particles, after crossing the Turkish mainland over night, is transported over the AS, with most of its air masses above the stable MABL. The plume is

moving fast with rather negligible mixing, especially above the MABL, thereby affecting areas located further away, such as the central AS, within a couple of hours after sunrise on 23 July (around 9:00 LST) and around noon on 24 July. The rapid advection above MABL, leaves almost intact the majority of the newly formed particles, formed upwind due to favorable chemical background conditions and solar irradiance. Thereafter, while the part of the plume above the MABL passes over the Cyclades complex, the wakes on the lee side of the islands enhances vertical mixing, enabling its entrainment into the

MABL. The freshly nucleated particles that remained constantly inside the well-mixed MABL, suffered an early ageing (i.e. growth by condensation and coagulation). The concentrations at both nucleation and Aitken modes jump almost simultaneously accompanied by concurrent increases in $O_3$, $NO_2$ and $SO_2$ concentrations, at Santorini station, during these two consequent events (Fig. 4). This could be an indication that this station receives masses simultaneously from different layers (inside and above the MABL), in line with a number of cases where maximum rate of change of ultra-fine particle

concentrations close to the surface was always preceded by breakdown of the nocturnal inversion and enhancement of vertical mixing (Crippa et al., 2012). Simulation results are in qualitative agreement with these findings, as the concentrations of Aitken mode (nucleation mode is assigned to the Aitken mode) are substantial at these two distinct layers. In particular, a plume with large particulate load in the Aitken mode ($> 1 \times 10^4$ particles cm$^{-3}$) is spread over northwestern Turkey and subsequently advances over the AS, both inside and above the MABL (Fig. S7 B and C). Our results agree with

previous observations during an Etesian event, where number concentrations up to ~$1.2 \times 10^4$ cm$^{-3}$ were observed at the northeastern AS with the Aitken-mode particles dominating by up to 70% (Tombrou et al., 2015). According to the simulations, the air masses passing over northwestern Turkey (area A in Fig. 1), are enriched with sufficient $H_2SO_4$ (Fig. 9) as well as with biogenic isoprene from the forested area (not shown) both necessary to initiate NPF. Due to limited concentrations of $NH_3$ in this area, according to the simulations (not shown), the air masses over northwestern Turkey are





less neutralized (the molar ratio of ammonium over inorganic ions concentrations is < 0.5; Fig. S7 A) compared to the more neutral aerosol reaching the western part of AS. This chemical background over northwestern Turkey seems to favor the intensity of the NPF events; the less intense event on 24 July is consistent with the much lower simulated concentrations in the Aitken mode over this area ($3.5 \times 10^3$ particles cm$^{-3}$) as well as with more neutral aerosols (molar ratio of ammonium over inorganic ions concentrations ~ 0.8-0.9).

The air masses arriving 3-h later (after 13:00 LST) at Finokalia, on 23 July (Fig. 4) have trace a lower number of nucleated particles, but higher number of Aitken-mode particles likely from the microphysical evolution of the NPF particles during their transport towards Crete. The 3-h transit timescale is in agreement with the prevailing wind speed (about 10 m s-1; Fig. S1) and the 120 km distance between Santorini and Finokalia. The simulated Aitken mode ranges from 4 to $5 \times 10^3$ cm$^{-3}$) peaking around 15:00 LST (data not shown). The differences between the two stations corroborate the importance of site-to-site variability even in cases of regional NPF events. Based on the current simultaneous measurements along the same flow stream, it seems that both stations are under the influence of regional NPF events, during the Etesians. Nevertheless, the nucleation mode particles are significantly reduced as they have shifted gradually towards larger sizes, before reaching Finokalia.

During the period of moderate winds on 26 July, the air masses arriving at lower levels (below 500 m agl) at Santorini station (Fig. 2) have mainly passed over continental areas, and have been substantially enriched by anthropogenic emissions, while those at higher levels have covered longer distances over Eastern Europe at the same time. These atmospheric conditions stimulate the mixing of air masses with local anthropogenic and natural emissions favoring photochemical production of secondary pollutants such as $O_3$ (Figs. 4, S4) and secondary aerosols (Fig. 6) and probably limiting the NPF events. WRF-Chem simulations also provide additional evidence that high concentrations of gases (e.g. $O_3$ shown in Fig. 5) and secondary aerosols (e.g. $SO_4$ shown in Fig. 7) have been previously well mixed and neutralized (molar ratio of ammonium over inorganic ions concentrations equal to ~ 1) either over the eastern Balkans and/or the western part of Turkey before starting their journey over the AS.

### 3.5 Impact of NPF events to CCN production and droplet number

Understanding how NPF affects cloud formation requires quantification of its impact on the CCN levels that develop for cloud-relevant supersaturations. Although CCN concentrations were not measured, they can still be calculated using the observations of size distribution and chemical composition as follows. First, Köhler theory (Köhler, 1936; Seinfeld and Pandis, 2006; Petters and Kreidenweis, 2007) is applied to determine, based on knowledge of aerosol composition, the minimum dry size of particles, $d_c$, that can activate at a given level of supersaturation, $s$. Then, the CCN concentration is determined from the observed size distributions, by calculating the concentration of particles with sizes above $d_c$ (Seinfeld and Pandis, 2006). $s$ is either prescribed or determined from a cloud parameterization, both of which are used here. Chemical composition is expressed in terms of the hygroscopicity parameter, $\kappa$, (Petters and Kreidenweis, 2007). The presentation of the results and the relevant discussion will be based on the periods before and after the NPF events.





CCN concentrations are calculated for prescribed values of $s$ between 0.2 and 0.8%, corresponding to supersaturations found in relatively pristine stratiform to convective clouds (Seinfeld and Pandis, 2006). $\kappa$ is calculated for each time step from the $PM_1$ chemical composition observed at Finokalia as follows: $\kappa = \varepsilon_{inorg}\,\kappa_{inorg} + \varepsilon_{org}\,\kappa_{org}$, where $\kappa_{inorg} = 0.6$ is the value for ammonium sulfate (Petters and Kreidenweis, 2007), and $\kappa_{org} = 0.16$ corresponds to the organic fraction

(Bougiatioti et al., 2009), and $\varepsilon_{inorg}$, $\varepsilon_{org}$ are the volume fractions of each constituent measured at Finokalia. The volume fractions range from 0.45 to 0.76 for inorganics and from 0.24 to 0.55 for organics, similar to the values measured under comparable atmospheric conditions by Bougiatioti et al. (2009, 2011) and Bezantakos et al. (2013). Throughout the measurement period, the aerosol exhibited predicted values of hygroscopicity from 0.20 to 0.39, which is also consistent with the values determined by Bougiatioti et al. (2009, 2011) and Bezantakos et al. (2013). The aerosol hygroscopicity follows a

diurnal cycle being minimum just before noon and becoming maximum late in the afternoon, owing to a higher sulfate-to-organic mass ratios (Fig. 6). Consequently, average $\kappa$ values were estimated to be higher after the NPF events compared to the period before (increase by ~35% on 23 July and up to 15% on 24 July). Given a lack of $PM_1$ chemical composition measurements at Santorini, the chemical composition at Finokalia is applied instead to the Santorini size-distribution observations. The WRF simulations support this assumption, as a similar chemical behavior is simulated for both stations

(Figs. 5,7). The model systematically underestimates the organic fraction at both stations (organic volume fraction does not exceed 0.2), but with minimal impact on resulting $\kappa$ values, as they do not differ from measurements for more than ±6% throughout the simulation period. From long-term measurements in the study area, the relative contribution of the main $PM_1$ constituents, including ammonium, is quite consistent over the years (Sciare et al., 2003; Kouloouri et al., 2008, Bougiatioti et al., 2013). Thus, a sensitivity test of CCN concentration at Santorini to shifts in $\kappa$ by ±20%, is also carried out.

The resulting CCN timeseries during Etesian flow are shown in Figure 10. Average values of $\kappa$, $d_c$, and CCN concentrations at both stations, before and after the NPF events are provided in Table 3. The calculated CCN number concentrations follow a diurnal cycle and tend to be maximum during the afternoon, after the NPF events, following the increase of $\kappa$ values. Most particles are CCN-active for $s \geq 0.6\%$, as they converge towards the total CN time series. Bougiatioti et al. (2009) observed similar behavior at Finokalia for polluted air masses with a similar origin (Balkans). For $s = 0.6\%$, $d_c$

varies from 43 to 51 nm and CCN concentrations reach up to $\sim 6 \times 10^3$ cm$^{-3}$ following the Aitken-mode concentrations at both stations (Figs. 4, 10). The higher CCN number concentrations at Finokalia, compared to those observed at Santorini (Table 3), is due to the higher number of accumulation-mode particles passing previously from Santorini (that are too small to activate at Santorini, but have grown to CCN-relevant sizes by the time they arrive at Finokalia, section 3.3). Accordingly, the higher activation fractions (CCN/CN) are observed at the station of Finokalia, with larger and more aged aerosol particles, while at

Santorini this is observed at the end of the events, when the smaller particles drop in concentration because they grow to larger sizes. On 23 July, the NPF event increases the CCN concentrations by 157% at Santorini and 106% at Finokalia, compared to their pre-event values; while in some moments the increase can reach up to a factor of 6. During the second less intense event, on 24 July, the CCN increase is lower at both stations (31% at Santorini and 53% at Finokalia). The lower increase is also due





to the pre-event background, characterized by higher CCN concentrations. Changes in chemical composition, as described above exhibit a relative low variation in CCN concentrations (at $s = 0.6\%$) up to 10%, following the same diurnal behavior. As expected, lowering the supersaturation at 0.2% leads to the activation of larger particles with $d_c$ ranging from 91 to 106 nm that is consistent with the observations reported by Kalivitis et al. (2015). At $s = 0.2\%$, both NPF events contribute up to 50%

to the increase of the CCN concentrations at both stations. However, the higher CCN production at Finokalia on 24 July is associated with the accumulation-mode particles at the end of both events.

        Studying the impact of NPF on CCN concentrations for prescribed levels of supersaturation (the usual approach for observational studies of NPF) provides an incomplete picture, as it does not consider the feedback of CCN on cloud supersaturation that develops in cloudy updrafts. Mechanistic cloud droplet formation parameterizations (Ghan et al., 2011;

Morales Betancourt and Nenes, 2014) can capture this complexity by efficiently calculating the maximum supersaturation ($s_{max}$) that forms in a cloud given knowledge of the aerosol size distribution, composition and updraft velocity; the droplet number ($N_d$) that forms is then given by the CCN concentration at $s_{max}$. Using this approach, we calculate the droplet number and supersaturation for clouds forming in the vicinity of both sites during all NPF events. The droplet parameterization used is based on the "population splitting concept" of Nenes and Seinfeld (2003), later improved by Barahona et al. (2010) and

Morales Betancourt and Nenes (2014). In the calculations of droplet number, the size distribution is represented by the sectional approach, derived directly from the SMPS distribution files. The updraft velocity has been calculated from high-resolution airborne measurements performed over this region of AS, under similar atmospheric conditions (Tombrou et al., 2015). The observations suggest that the distribution of vertical velocities in the boundary layer display a spectral dispersion of $\sigma_w = 0.2\text{-}0.3$ m s$^{-1}$ around a zero average value, which is consistent with vertical velocities observed in marine boundary layers

(e.g., Meskhidze et al., 2005; Ghan et al., 2011). When applying the droplet parameterization, we employ the "characteristic velocity" approach of Morales and Nenes (2010) to obtain velocity PDF-averaged values of cloud droplet number concentration (CDNC) and $s_{max}$. As a sensitivity test, we also consider calculations for a convective boundary layer ($\sigma_w = 0.6$ m s$^{-1}$).

        The calculation of PDF-averaged values of $N_d$ and $s_{max}$ is carried out for every distribution of aerosol number and

composition measured for all NPF events. The calculated timeseries are shown in Figure 11 for Santorini (top panel) and Finokalia (bottom panel). $s_{max}$ is negatively correlated with $N_d$ at both stations, owing to the increased competition for water vapor by the growing droplets when CCN increase. As a result, $N_d$ responds sublinearly to CCN increases – the degree to which this depends on the level of aerosol concentrations before and during the NPF event. At Santorini, the CCN levels are much lower than at Finokalia (Table 3), so we expect the relative increase in $N_d$ from NPF to be higher there. Assuming $\sigma_w =$

0.3 m s$^{-1}$, the NPF events are associated with $s_{max}$ decreases at both stations, compared to the period before the events. On 23 July, the decrease is on average 12% at Santorini and 9% at Finokalia. As a result, $N_d$ concentrations during the NPF event increases by 13% to 124±8 cm$^{-3}$ at Santorini, compared to the period before the event. At Finokalia, however, aerosol levels are much higher and $N_d$ remains virtually the same before and after the NPF event (Table 3). The effect of the less intense NPF





event on 24 July is higher; $N_d$ concentration increases by 36% at Santorini and 4% at Finokalia compared to pre-event values. The decrease of $s_{max}$ is also higher on this day, 17% at Santorini and even higher 36.4% (at 0.06-0.08%) at Finokalia (Table 3) owing to the higher accumulation particle concentrations compared to the previous events. The variance of $N_d$ during the event period, for $\sigma_w$ equal to 0.3 m s$^{-1}$, is 475 cm$^{-3}$ at Santorini and 37 cm$^{-3}$ at Finokalia, while for $\sigma_w$ equal to 0.6 m s$^{-1}$ the variance

is 865 cm$^{-3}$ and 20 cm$^{-3}$, respectively. Altogether, this clearly shows that when NPF particles age (e.g., arrive at Finokalia) their competition for water vapor can reduce cloud supersaturation to very low levels.

        The larger updraft velocity ends in larger values of $s_{max}$, which allow smaller particles to activate into cloud droplets. In particular, $N_d$ exhibits a substantial increase for $\sigma_w = 0.6$ m s$^{-1}$, but with a similar pattern to that with the lower velocity, especially at Santorini indicating that the impact of mean vertical velocity on the CDNC is higher at this station. In this case,

the average $N_d$ concentration is 217±15 cm$^{-3}$ at Santorini and 619±109 cm$^{-3}$ at Finokalia (increase relative to $\sigma_w = 0.3$ m s$^{-1}$ by 75% and 52%), after the event on 23 July and 286±15 and 786±11, respectively (increase relative to $\sigma_w = 0.3$ m s$^{-1}$ by ~76%, for both stations), on 24 July. It is interesting to note that for $\sigma_w = 0.6$ m s$^{-1}$ two $N_d$ peaks are observed at Finokalia, from which the first is attributed to local processes as it is observed much earlier than the NPF event at Santorini. The stronger variation of $N_d$ at Finokalia, under the higher vertical wind, compared to Santorini, indicates that vertical velocity variations likely

dominate the variance of droplet number for clouds in the region of Finokalia. Furthermore, from the partial sensitivity of $N_d$ to the total aerosol number, and to $\kappa$, the relative contribution of chemical composition and total aerosol number to the variance of $N_d$ is attributed. We find that in most cases the predicted $N_d$ variability is almost exclusively governed by the aerosol number variation ($> 98\%$, Table 3) and to a lesser extent by the chemical composition ($< 2\%$). The relative contribution of chemical composition becomes more significant at Finokalia only after the intense NPF event on 23 July (10%

for $\sigma_w = 0.3$ m s$^{-1}$ and 19% for $\sigma_w = 0.6$ m s$^{-1}$). This can be attributed to the more "aged" nature of the sampled aerosol at Finokalia, compared to the one at Santorini. This is consistent with the lower $s_{max}$ observed at Finokalia, leading to the activation of larger particle sizes that have been subject to longer atmospheric processing, during their transition to more unstable conditions after Santorini. Altogether, although NPF events may strongly elevate CCN numbers, the relative impacts on cloud droplet number (compared to pre-event levels) is eventually limited by water vapor availability and depends on the

aerosol levels associated with the background.

## 4 Conclusions

Concentrations of chemically- and size-resolved submicron aerosol particles along with concentrations of trace gases and meteorological variables have been simultaneously measured at Santorini (central AS) and Finokalia on Crete (southern AS) from 15 to 28 July 2013. Two well-distinguished periods are identified: the first with strong wind speeds and wind directions

forming the characteristic 'ring-shape' of the Etesian flow around Turkey, and the second with moderate wind speeds and northerly direction over the AS. The two periods exhibited intense differences on air quality levels.



In general the inorganic and organic aerosol mass concentrations have a similar to ozone behavior during most of the experimental period. During Etesian conditions, the mass concentrations were reduced roughly by a factor of two compared to those during the moderate wind period. The total number concentration of aerosol particles was increased during the Etesian flow, varying from $1.5 \times 10^3$ to $1.5 \times 10^4$ particles cm$^{-3}$ at Santorini and from $2.4 \times 10^3$ to $7.5 \times 10^3$ particles cm$^{-3}$ at Finokalia. Furthermore, intense burst of nucleation-mode particles have been recorded at both stations, with more intense those observed at Santorini. At Finokalia, the fragment of nucleated particles is diminished, and a higher number concentration of the Aitken-mode particles is observed, attributed to downward mixing and photochemistry. The nucleation mode particles are gradually shifting towards larger sizes at both stations, however, at Santorini the number of particles remains high for several hours, indicating regional NPF. During the period of moderate winds, the total number concentration of the particles reaches lower values, while nucleation-mode particles are not detected at any of the stations.

The observed NPF events have been initiated at least 250 km (covered within 4.5 hours) to the northeast of Santorini in the center of AS, upwind of Cyclades complex, under favorable meteorological conditions, under a strong-channeled northeastern wind flow received by both stations. The plume, after crossing Turkish mainland during the night, is transported over the AS, with most of its air masses remaining above the stable MABL. The fast advection above MABL leaves intact most of the newly formed particles formed upwind, despite that the wakes on the lee side of the islands enhance vertical mixing, enabling its subsequent entrainment into the MABL. The freshly nucleated particles that remained constantly inside the well-mixed MABL, suffered an early ageing (i.e. growth by condensation and coagulation).

To understand the impact of NPF on CCN levels, using the $\kappa$ of particles and in conjunction with a typical supersaturation for the area, we calculated the number concentration of particles which act as CCN at both stations. It occurred that due to NPF, CCN concentrations augment considerably during early afternoon (87% on average for both stations and both events), with concentration levels at Finokalia being higher due to particle growth and atmospheric processing. Calculations of droplet number that would form in clouds influenced by the observed airmasses indicate that NPF augments droplet number, but to a much lesser extent (12%) than implied by CCN variation. This behavior demonstrates there is a limit in the amount of droplets that NPF can contribute because the supersaturation in cloud depresses (here, by roughly 14%) as additional CCN are added from NPF. The pre-NPF aerosol levels and prevailing dynamics of the clouds determine the degree of water vapor competition and precondition clouds to be sensitive - or not - to further CCN increases from NPF.

**Acknowledgments**

AN acknowledges support from a Georgia Power Faculty Scholar Chair and a Cullen-Peck Faculty Fellowship.



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





**Tables**

**Table 1.** Summary of the variables and operation characteristics of the instruments at Santorini and Finokalia stations.

| Santorini | Instrument | Resolution | Period of Operation |
|---|---|---|---|
| *Aerosols* | | | |
| Aerosol Number Distribution (10 - 500 nm) | TSI 3034 SMPS | 3 min | 15 – 28 July |
| *Gaseous Species* | | | |
| $O_3$ | M400E Photometric ozone analyzer | 1 min | 18 – 28 July |
| $SO_2$ | M100E UV Fluorescence analyzer | 1 min | 15 – 28 July |
| NO, $NO_2$, $NO_x$ | M200E Nitrogen Oxide analyzer | 1 min | 15 – 28 July |
| **Finokalia** | | | |
| *Aerosols* | | | |
| Aerosol Number Distribution (9 – 848 nm) | TROPOS type SMPS | 5 min | 16 – 29 July |
| Chemical composition ( $SO_4^{2-}$, $NO_3^-$, $NH_4^+$, $Cl^-$, organics) | Aerodyne Research Inc. Aerosol Chemical Speciation Monitor (ACSM) | 30 min | 15 – 28 July |
| *Gaseous Species* | | | |
| $O_3$ | Thermo electron Model 49I | 3 min | 15 – 28 July |
| *Meteorology* | | | |
| Relative humidity, Temperature | MP101A Humidity-Temperature | 5 min | 15 – 28 July |
| Wind Speed, direction | 05103 Wind Monitor | 5 min | 15 – 28 July |





**Table 2.** Average (± standard deviation) of $O_3$ concentrations and aerosol mass and number concentrations during two periods, from 22 to 24 July (Etesian flow) and from 25 to 27 July (moderate northerly winds).

| Tracer | Santorini | Finokalia | Santorini | Finokalia |
|---|---|---|---|---|
| | Period 1: 22/7 - 24/7 | | Period 2: 25/7 - 27/7 | |
| $O_3$ (ppbv) | $51.4 \pm 6.2$ | $53.8 \pm 4.5$ | $70.0 \pm 8.5$ | $71.0 \pm 7.9$ |
| Sulfate ($\mu g\ m^{-3}$) | N/A | $3.1 \pm 1.2$ | N/A | $7.3 \pm 1.5$ |
| Ammonium ($\mu g\ m^{-3}$) | N/A | $1.4 \pm 0.6$ | N/A | $3.1 \pm 0.6$ |
| Organics ($\mu g\ m^{-3}$) | N/A | $4.2 \pm 1.3$ | N/A | $8.6 \pm 1.2$ |
| Nitrate ($\mu g\ m^{-3}$) | N/A | $0.38 \pm 0.12$ | N/A | $0.8 \pm 0.1$ |
| Total number conc. ($cm^{-3}$) | $3.6 \pm 2.1 \times 10^3$ | $3.9 \pm 1.2 \times 10^3$ | $2.0 \pm 0.6 \times 10^3$ | $3.6 \pm 0.5 \times 10^3$ |
| Aitken mode ($cm^{-3}$) | $2.2 \pm 1.4 \times 10^3$ | $2.5 \pm 1.5 \times 10^3$ | $1.2 \pm 0.5 \times 10^3$ | $1.6 \pm 0.5 \times 10^3$ |
| Accumulation mode ($cm^{-3}$) | $9.6 \pm 3.5 \times 10^2$ | $1.6 \pm 0.9 \times 10^3$ | $1.0 \pm 0.5 \times 10^3$ | $2.1 \pm 0.6 \times 10^3$ |



**Table 3**: Average (± standard deviation) of calculated $\kappa$ using the PM$_1$ chemical composition at Finokalia, the $d_c$ (calculated according to Petters and Kreidenweis, 2007), and the estimated CCN concentration particles at both stations, on 23 and 24 July. Here $s_{max}$ is the maximum supersaturation in the cloud, N$_{total}$ is the total particle number concentration, and $N_d$ is the potential cloud droplet number concentration calculated according on the parameterizations provided by Morales and Nenes (2010), and by using the probability density function (PDF) of the characteristic updraft velocity ($\sigma_w$ =0.3 m s$^{-1}$ and $\sigma_w$ =0.6 m s$^{-1}$). Time is in LST

| | Santorini | | Finokalia | | Santorini | | Finokalia | |
|---|---|---|---|---|---|---|---|---|
| | 23/7 | | | | 24/7 | | | |
| | *Before* 00:00 - 8:00 | *After* 15:00 – 21:00 | *Before* 00:00 - 10:00 | *After* 17:00 – 21:00 | *Before* 00:00 - 10:00 | *After* 18:00 – 21:00 | *Before* 00:00 - 11:00 | *After* 17:00 – 21:00 |
| $\kappa$ | 0.287±0.014 | 0.355±0.026 | 0.283±0.020 | 0.381±0.017 | 0.296±0.011 | 0.337±0.012 | 0.299±0.014 | 0.344±0.007 |
| $d_c$ (nm) ($s$=0.2%) | 103.6±1.8 | 95.1±2.2 | 103.8±2.1 | 93.9±1.3 | 101.3±1.4 | 95.9±1.2 | 101.7±1.7 | 96.6±0.6 |
| $d_c$ (nm) ($s$=0.6%) | 49.8±0.9 | 45.7±1.1 | 49.9±1.0 | 45.1±0.6 | 48.7±0.7 | 46.1±0.6 | 48.8±0.8 | 46.4±0.3 |
| CCN$_{0.2}$ (cm$^{-3}$) | 536±27 | 794±145 | 1002±76 | 1420±383 | 682±66 | 1028±61 | 1062±156 | 1822±154 |
| CCN$_{0.6}$ (cm$^{-3}$) | 1225±90 | 3155±789 | 2111±196 | 4343±1119 | 1535±66 | 2004±224 | 2191±270 | 3346±399 |
| N$_{total}$ (cm$^{-3}$) | 1777±421 | 4621±1986 | 3506±699 | 5710±779 | 2306±154 | 2557±351 | 3198±384 | 3921±404 |
| *$\sigma_w$=0.3 m s$^{-1}$* | | | | | | | | |
| $s_{max}$ (%) | 0.25±0.01 | 0.22±0.01 | 0.11±0.01 | 0.10±0.01 | 0.23±0.01 | 0.19±0.01 | 0.11±0.01 | 0.07±0.01 |
| $N_d$ (cm$^{-3}$) | 110±4 | 124±8 | 423±4 | 407±19 | 121±5 | 165±9 | 423±3 | 440±5 |
| Activation Fr. (%) | 6.5±1.5 | 3.1±1.1 | 12.5±2.4 | 7.2±0.6 | 6.4±0.4 | 7.9±1.2 | 13.4±1.6 | 11.3±1.1 |
| Contribution of $\kappa$ (%) | 1.4 | 2.7 | 2.6 | 10.2 | 0.7 | 1.9 | 0.9 | 0.3 |
| Contribution of $N_{aer}$ (%) | 98.6 | 97.3 | 97.4 | 89.8 | 99.3 | 98.1 | 99.1 | 99.7 |
| *$\sigma_w$=0.6 m s$^{-1}$* | | | | | | | | |
| $s_{max}$ (%) | 0.32±0.01 | 0.28±0.01 | 0.14±0.01 | 0.14±0.01 | 0.29±0.01 | 0.23±0.01 | 0.14±0.01 | 0.11±0.01 |
| $N_d$ (cm$^{-3}$) | 192±6 | 217±15 | 627±67 | 619±109 | 213±7 | 286±15 | 621±73 | 786±11 |
| Activation Fr. (%) | 11.4±2.6 | 5.4±1.9 | 18.8±5.1 | 10.8±0.7 | 11.3±0.7 | 13.7±2.1 | 19.7±3.1 | 20.2±1.9 |
| Contribution of $\kappa$ (%) | 1.2 | 1.9 | 3.8 | 19.0 | 0.6 | 1.6 | 0.7 | 0.2 |
| Contribution of $N_{aer}$ (%) | 98.8 | 98.1 | 96.2 | 81.0 | 99.4 | 98.4 | 99.3 | 99.8 |





**Figure captions**

**Fig. 1**. The extended area of study. The stations of Santorini and Finokalia are also indicated.

Fig. 2. HYSPLIT4 back-trajectories computed with an end point at the Santorini station, on 23 (left panel), 24 (central panel) and 26 July (right panel), 2013. Single back trajectories from the heights of 100, 500 and 1000 m (upper panel).

**Fig. 3.** Time series of the wind speeds (solid lines, left axis) and wind directions (open cycles, right axis) at Santorini (simulations by the WRF-Chem model) and at Finokalia (measurements). The second period of the Etesian flow is shaded with yellow and the

period of moderate northerly flow with grey

**Fig. 4**. Aerosol modal number concentrations, meteorological and pollutant concentrations at Santorini (top panel) and Finokalia (bottom panel). Note that $SO_2$ is shown at Santorini, while $SO_4$ is shown for Finokalia.

**Fig. 5**. Spatial distribution of $O_3$ concentration (ppb) and wind speed at 400 m asl over the extended area of Greece as simulated by WRF-Chem at 15:00 LST for 23 July (left panel), and 26 July (right panel).

**Fig. 6.** Mass concentrations of submicron aerosol measured at Finokalia station.

**Fig. 7**. As in Fig. 4, but for sulfate concentration ($\mu g\ m^{-3}$) at 09:00 LST for 23 July (left panel), and 26 July (right panel).

**Fig. 8**. Diurnal evolution of the aerosol size-distribution on 23 and 24 July at Santorini (top panel) and Finokalia (bottom panel). The white dots stand for nucleation, the black dots for Aitken and the purple dots for accumulation geometric mean diameter.

**Fig. 9**. Spatial distribution of $H_2SO_4$ concentration and wind at 400 m asl over the extended area of Greece as simulated by WRF-Chem; at 06:00 LST (left panel) and 09:00 LST (right panel), on 23 July 2013.

**Fig. 10**. Time series of the CN and estimated CCN concentration particles, for various supersaturations, at Santorini (top panel) and Finokalia (bottom panel), during 23 and 24 July. Time is in LST.

**Fig. 11.** Time series of the estimated cloud droplet number concentrations ($N_d$), and maximum supersaturation in the cloud ($s_{max}$) for updraft velocities ($\sigma_w$) of 0.3 m s$^{-1}$ and 0.6 m s$^{-1}$, at Santorini (top panel) and Finokalia (bottom panel), during 23 and 24 July. Blue and red lines correspond to updraft velocity ($\sigma_w$) equal to 0.3 m s$^{-1}$, while orange and green to 0.6 m s$^{-1}$.




**Figures**

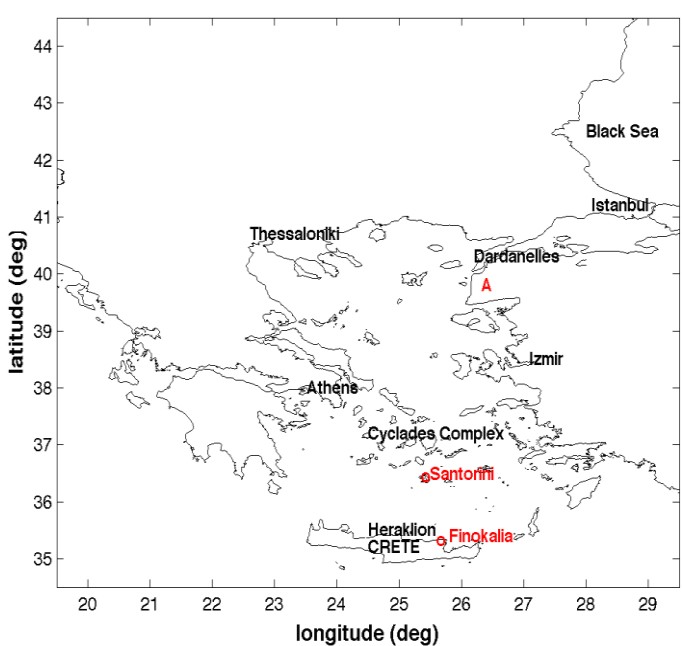

**Fig. 1**

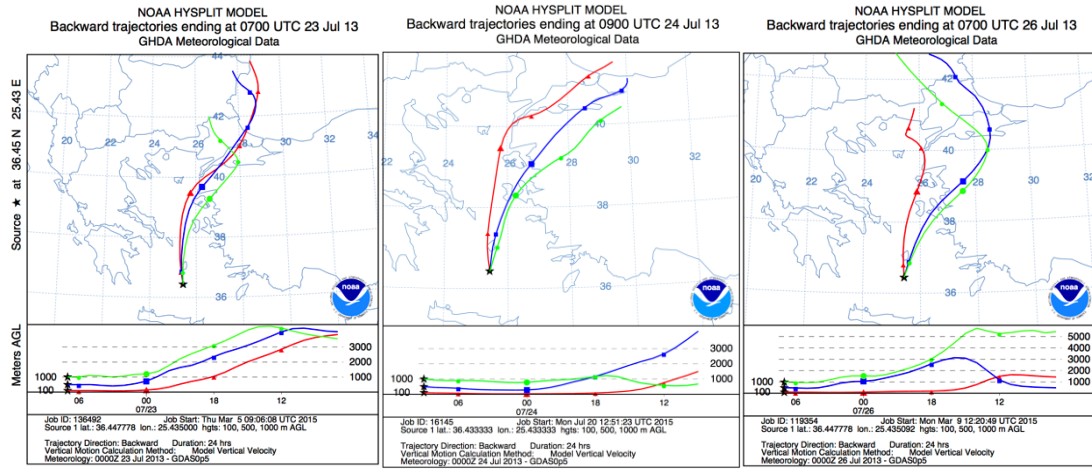

**Fig. 2**



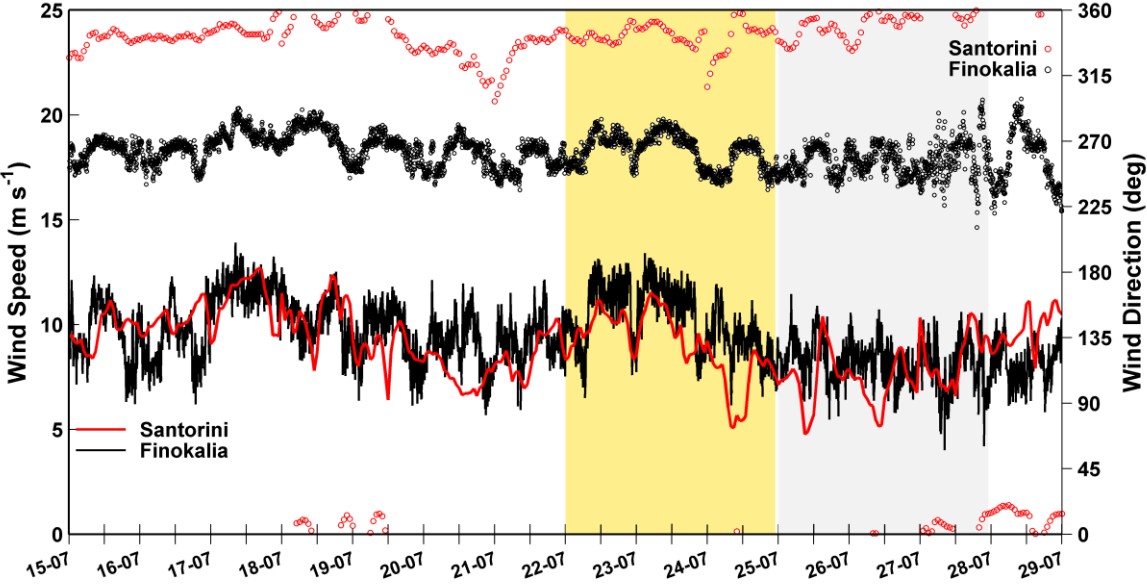

**Fig. 3**



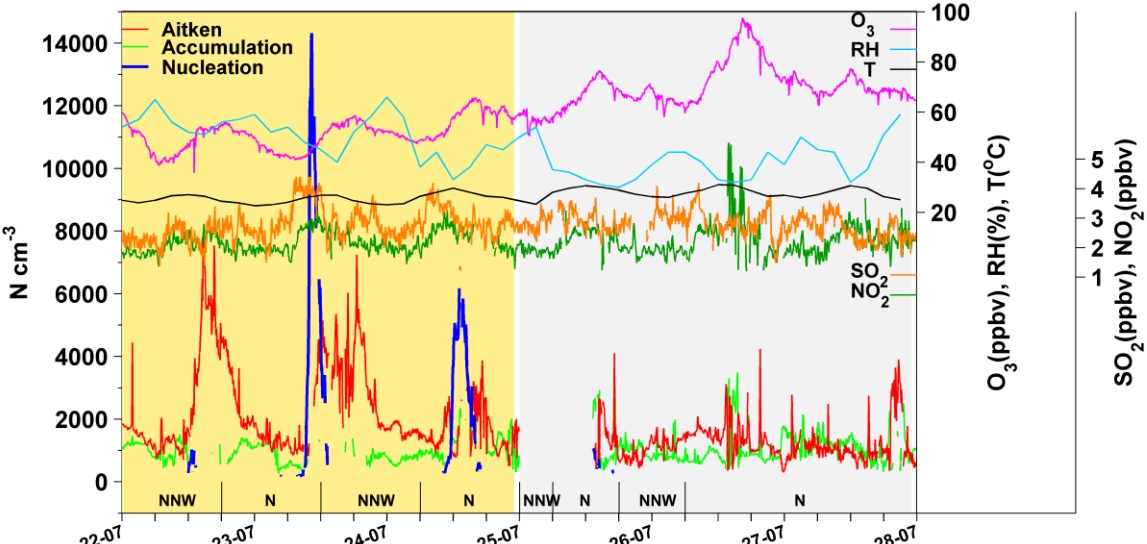

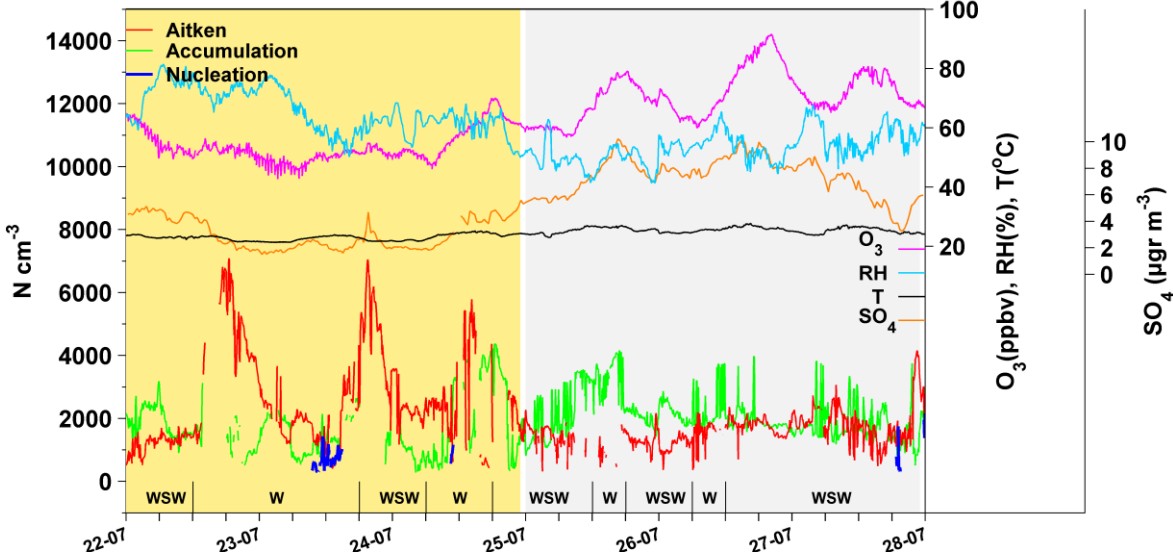

**Fig. 4**





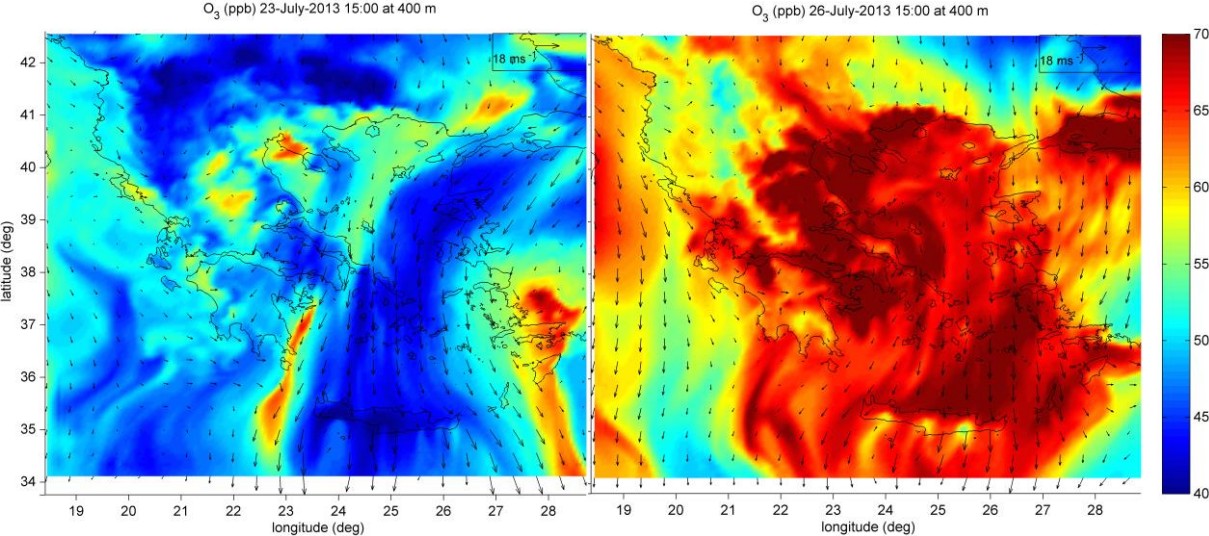

**Fig. 5**

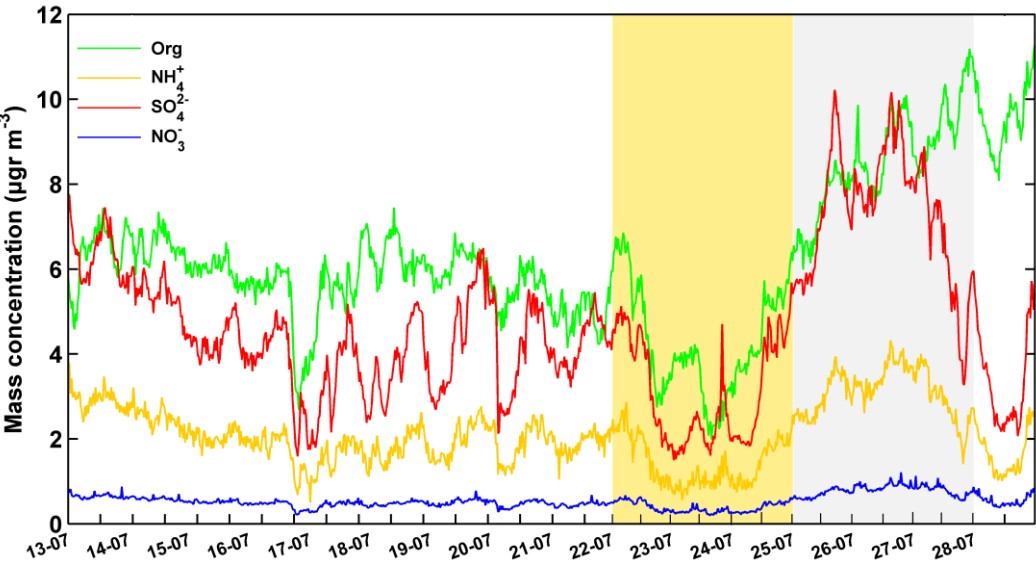

5   **Fig. 6**





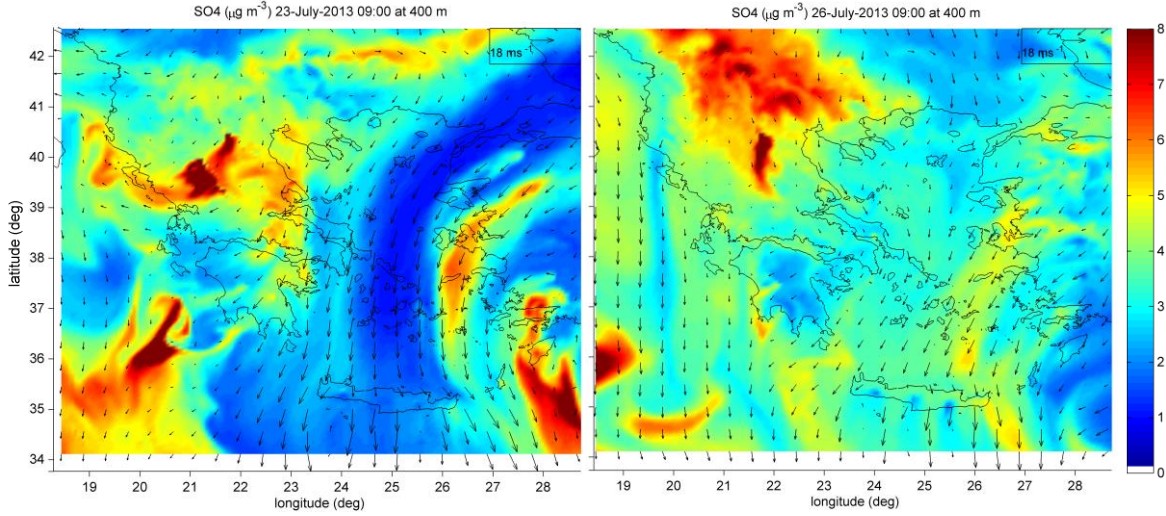

**Fig. 7**





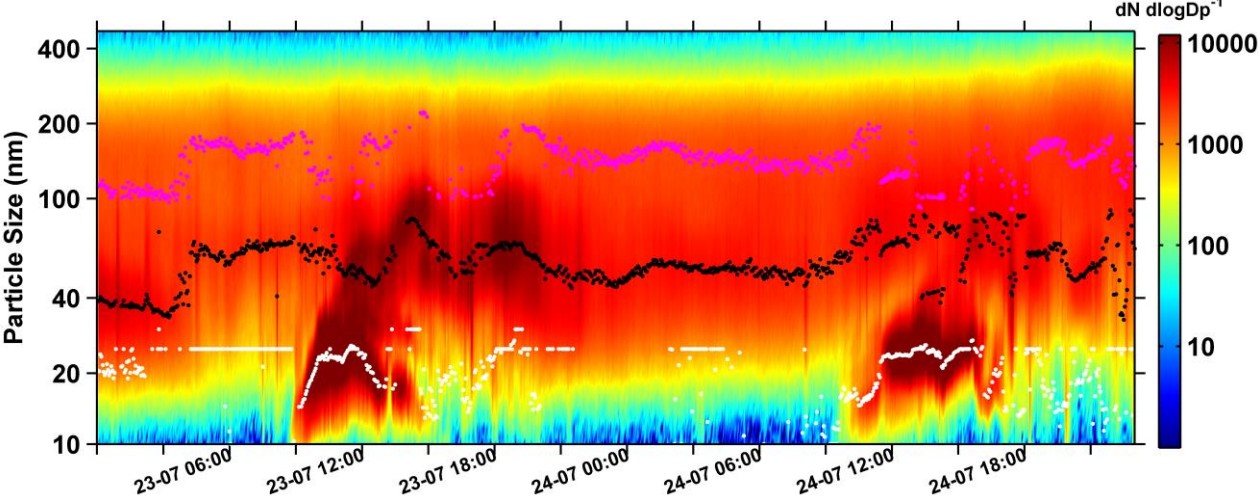

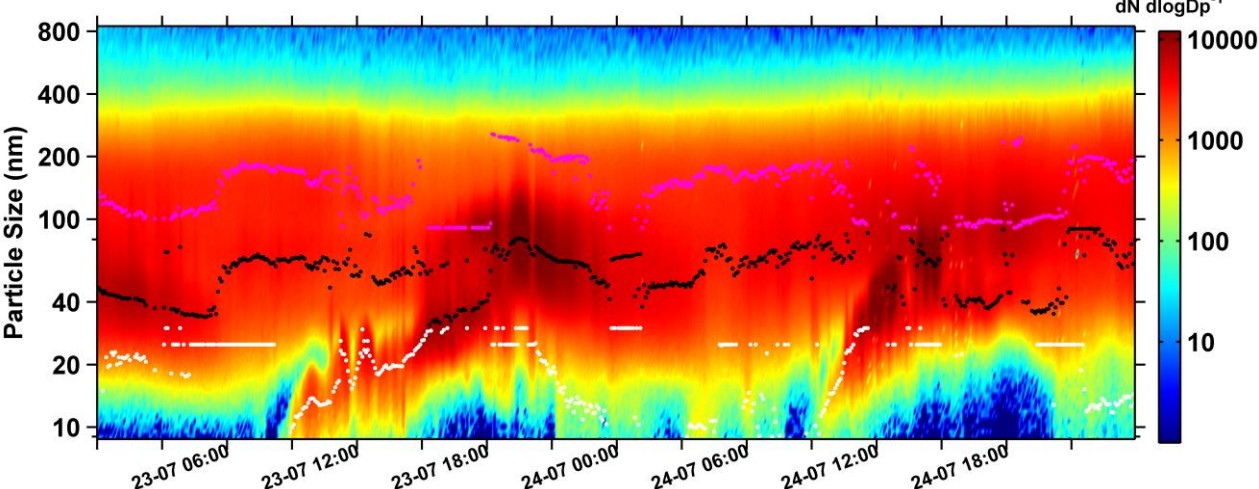

5   **Fig. 8**





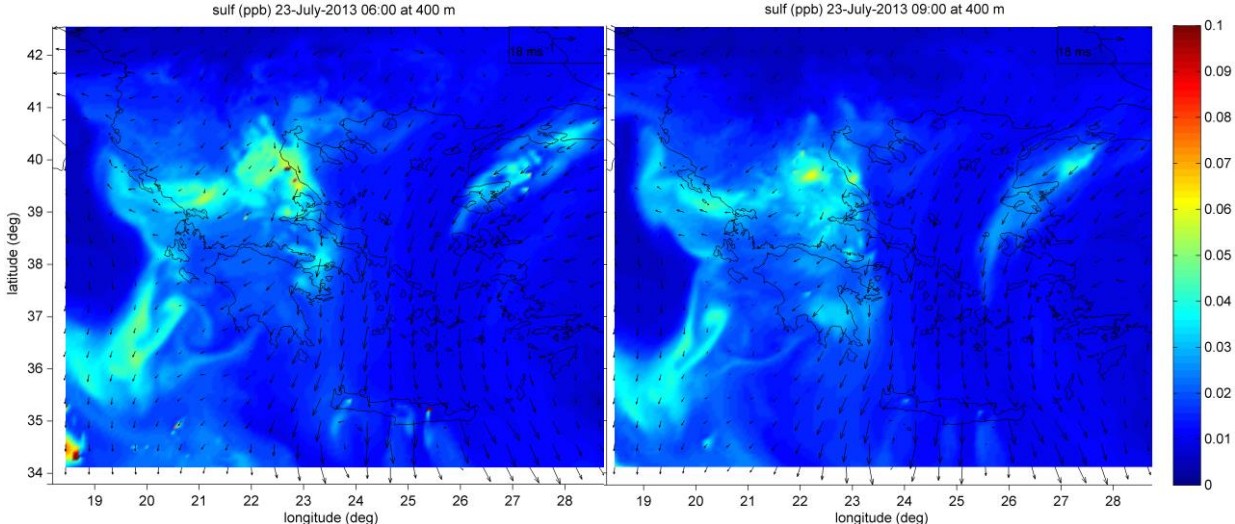

**Fig. 9**





**Fig. 10**





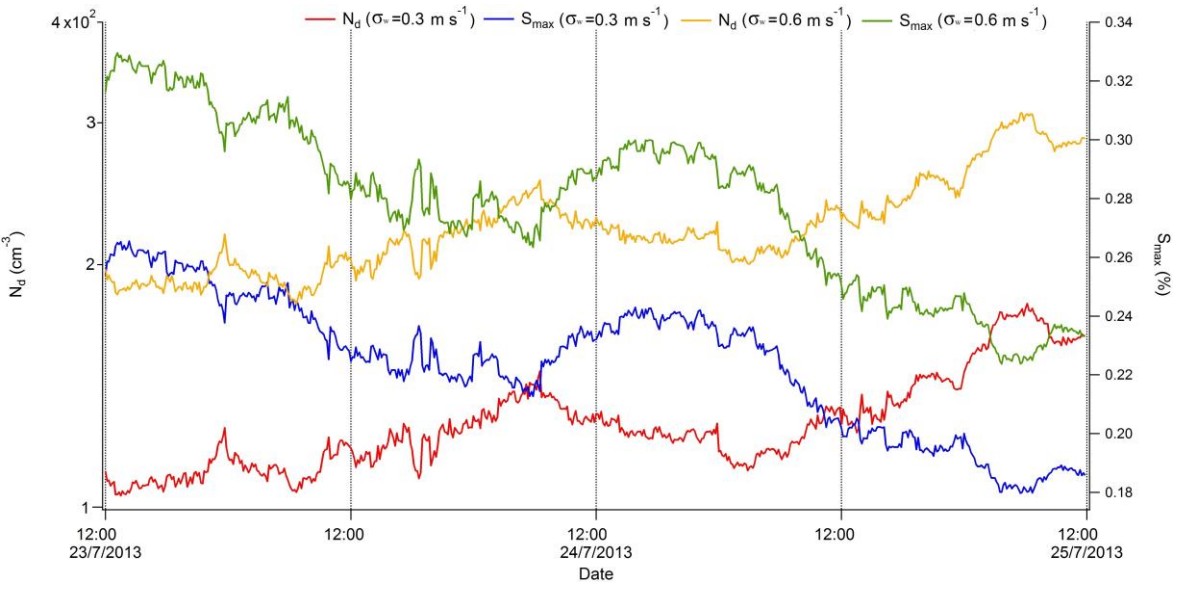

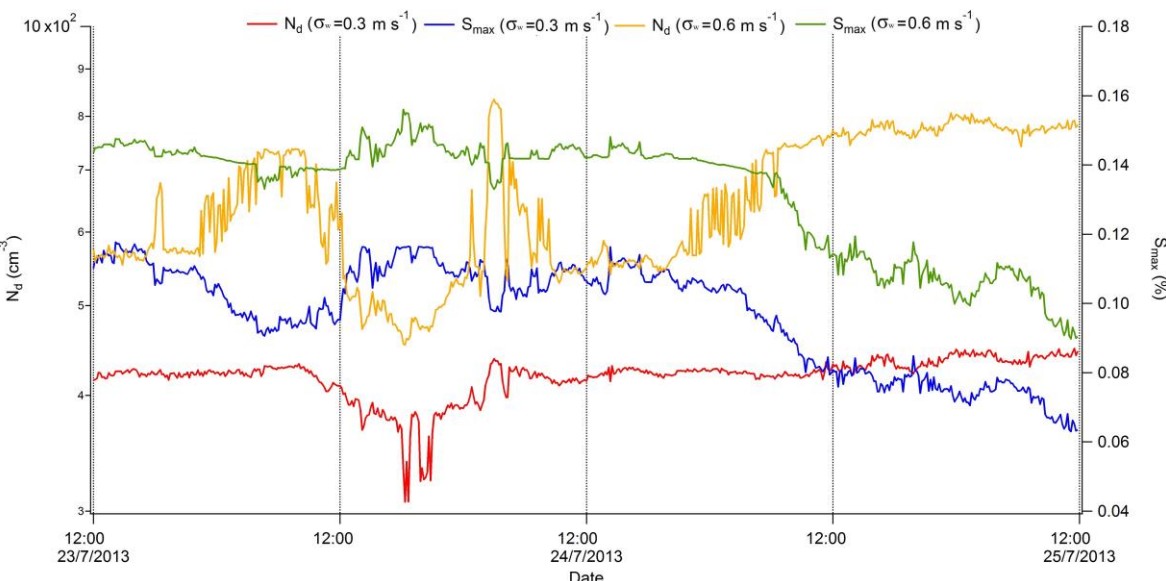

**Fig. 11**

