# Peer review of "New Particle Formation in the South Aegean Sea during the Etesians: importance for CCN production and cloud droplet number"

_Atmospheric Chemistry and Physics, 2016_

## Referee Comment (RC1) · Anonymous Referee #1 · 8 Aug 2016

General Comments:

Kalkavouras and co-authors present results from an intriguing experiment in the Aegean Sea. The nature of pollution arriving at the long-standing Finokalia measurement platform is investigated directly with observations at Santorini, a site strategically located along the trajectory from the European mainland. The observations target aerosol size distributions, relevant primary and secondary pollutant concentrations, and detection of new particle formation events. The authors identify two events representing characteristic flow from the north, albeit of two distinct types. The Etesian flow example is marked with NPF events at both sites, although the events are stronger at the Santorini site. The authors extend their observations of particle number and

composition to predictions of CCN at various supersaturations. They also go further to predict the total effect of the NPF events on cloud droplet number, taking into account the impact of constrained available water vapor. The paper starts with a nice scientific idea and goes into good detail into the results. What I see lacking is a little more connective tissue linking the observations at each site with each other and the mainland into a cohesive story. The material is already there, but it is somewhat buried and could be highlighted with a figure, for instance. I would like to see the following points addressed before recommending publication:

Specific Comments:

1. I recommend the authors adopt two shorthand names for the distinct periods (22/7-24/7 and 25/7-27/7). They could be referred to as "Etesian Flow" and "Moderate Surface Flow," for example. Small changes like this could help the readability of the paper. A useful addition to this paper would be a two-panel cartoon, each overlaid on the Greek domain map in Fig. 1, for example, that describes the factors at play in these two periods. They could identify generally where they expect emissions, mixing, oxidation, NPF, and aging of new particles to be happening.

2. The WRF-Chem aerosol module configuration, as documented by the authors, is problematic for this particular application. It is quite likely that NPF events and subsequent processing are not captured realistically at all by the model. The sulfuric acid/water pathway parameterized by Kulmala et al. (1998) is likely not strong enough to enhance particles near the surface and lower troposphere to the levels observed at the Santorini site. It is now well-documented that other reagents play important roles in this process (e.g. NH3 and organics), and these pollutants have been identified by the authors to be present and significant components of the aerosol. My guess is that most of the Aitken-mode particles in the model originate from direct emissions, not from secondary generation. A related issue is the lack of a dedicated nucleation mode in the model. Without this mode, any NPF events will artificially broaden the Aitken mode distribution and give unrealistic lifetimes against deposition and coagulation. It

will also affect the growth rates predicted by the model.

The authors astutely sidestep relying on the model to predict size distributions and use their own observations when possible for calculating CCN and cloud drop number concentrations. However, since they include an entire section (2.2) detailing the regional modeling they performed, it is a good idea to explicitly state the limitations of this analysis for particle size distributions, and remind the reader that they are using the WRF-Chem output for its knowledge of advection flows and chemical composition, not microphysical processing.

3. The authors conclude that the NPF events observed at Santorini are regional in nature with a spatial scale of 250 km and characteristic transport time of 4.5 hours. They also assert that Finokalia does not see the bursts because it is 3 hours away and particle populations age before they arrive there. The authors do note that this second observation demonstrates how site-to-site variability can be important during a regional event. I am not sure that this totally addresses the issue though. Why are the events sort of regional and sort of not-regional? Is this an issue of using up the NPF precursors before the air mass gets to Santorini and then shifting to chemical conditions that favor condensation to available surface area? Or is something else at play here?

4. The paragraph beginning line 13 on page 9 describes an interesting hypothesis for how pollutants are transported to the middle of the Aegean Sea with limited aging. However, I'm not convinced there is enough evidence to warrant the detailed discussion that is given to this possibility or the certainty with which it is treated in the conclusions section. As described in my first comment, any model data related to the size distribution of Aitken-mode particles probably cannot be trusted in this case. If I understand correctly, the main assertion here is that the particles were formed over the Turkish mainland and transported quickly before they have a chance to be significantly coagulated away. Why could the enhanced number concentrations not come instead from oxidation and NPF over the water during transport, where there may be enhanced photochemistry, complex interactions with clouds, interesting boundary layer phenomena, etc? If I'm not understanding the meat of the argument correctly, please explain it to me and consider rephrasing it in the text to be clearer. What insight do the model CO concentrations help to provide regarding the stratification and mixing of distinct layers downwind of the continent?

5. I recommend separating the paragraphs detailing the Nd calculations (starting on Page 12) into their own section, perhaps called "Impact of NPF events on cloud droplet number." Then section 3.5 would be called "Impact of NPF events on CCN production."

6. How is the partial sensitivity of cloud droplet number to chemical composition and vertical velocity determined? Can the equations be provided? What is the uncertainty associated with this? Please document it if possible.

Minor Changes/Typos:

Pg 2, Ln 25: The phrase "without any particular seasonal preference about their occurrence" is difficult to understand. Can the authors please reword this to be more specific?

Pg 3, Ln 4: "prior to reaching"

Pg 5, Ln 11-13: This is technically not a sentence.

Fig. 3 and caption: "open circles" not "cycles". Also, please indicate on the figure that the solid lines describe wind speed and the circles describe direction. It is hinted at in the figure and explained in the caption, but it would be quicker for the reader to have it identified visually, with an arrow or something.

Pg 6, Ln 16-17: The "less pronounced" diurnal cycle at Finokalia for ozone is not obvious to me from Fig. S4. Please include a plot of the actual diurnally averaged profiles or report the daily minima and maxima to demonstrate this point.

Pg 6, Ln 28-29: I would not characterize -21% or -15% under-prediction as "small".

Either establish what they are small compared to, or please get rid of this qualification.

Pg 6, Ln27-30: Please break this sentence up. It is long and confusing.

Pg 6, Ln 31-32: Is there a more recent or relevant reference than McKeen et al. (1991)? Anything that specifically identifies this model scenario or modern European scenarios in general as suffering from ozone boundary condition issues?

Pg 7, Ln 5-6: In what way did the inorganic and organic mass concentrations show "similar behavior" to that of ozone? Are the authors just identifying them all as secondary pollutants? Please provide an estimate of the correlation coefficient or index of agreement for a statement like this.

Pg 7, Ln 9: Please remove the comma after the parentheses.

Pg 7, Ln 20: Please refer to some quantitative statistics to back up this claim.

Pg 10, Ln 6-8: This sentence is worded in a confusing way.

Pg 10, Ln 25 – Pg 11, Ln 19: Most of this material would be better-placed in the methods section (2.3 maybe). This goes for the second paragraph a page 12 as well.

---

## Referee Comment (RC2) · Anonymous Referee #2 · 19 Aug 2016

The manuscript presents measurements of the number size distribution and chemical composition of submicron aerosols at two islands in the Eastern Mediterranean. The analysis is based on a measurement period over two weeks in the summer 2013, during persistent transport of continental airmasses from north to the sites. A chemical transport model and airmass back-trajectories are used to identify the source areas and transport routes of aerosols to the sites. Using case studies of two new particle formation (NPF) events the contribution of NPF to both the cloud condensation nuclei (CCN) and cloud droplet (Nd) concentrations is assessed. The results for CCN and Nd are based on Köhler theory and parameterizations.

I agree with the comments presented by the anonymous referee #1, and would like the

authors to address my further comments below. After addressing these comments I can recommend the manuscript for publication in Atmospheric Chemistry and Physics.

General comments:

Page 5, lines 6–7: Is it known what are the possible reasons for the underestimation of organic matter concentrations in the model results; could it be due to underestimation of primary emissions or underestimation of SOA formation in the model?

Page 5, lines 8–12: Care should be taken when using the HYSPLIT model with the GDAS 0.5° input data: the back-trajectory results might differ from those obtained with GDAS 1° input data due to the differences in the airmass vertical advection calculation method between these two datasets (see e.g. Su et al., 2015). Perhaps the authors could check that their back-trajectories shown in Figure 2 remain the same if using the GDAS with 1° resolution as input meteorological data.

Page 8, lines 6–7: Why are coagulation losses not included in the calculation of the formation rate of nucleation mode particles? This should be fairly straightforward to calculate based on the measured size distributions, and including the coagulation losses would make the calculated formation rates more readily comparable to literature values (which typically account for coagulation).

Page 10, line 4: Where does the 3 hour difference in the comparison between particle observations at Santorini and Finokalia come from? Based on the particle size distribution data in Fig. 8 the particle formation at both stations seems to start at 9 a.m. on 23 July, and the only appreacable difference in the particle concentrations in Fig. 4 seems to be in the nucleation mode concentration (i.e. intensity of particle formation).

Regarding the discussion on the CCN-sized particles and the calculated hygroscopicity parameters, it would be interesting to see how the results differ on days without new particle formation. This type of comparison between NPF and non-NPF days would put the results presented in the manuscript better into context with regard to the importance

of NPF to CCN and cloud droplet number at the Aegean Sea. Where there during the campaign any such non-NPF days for which the parameters of Table 3 could be calculated and reported for comparison with the two NPF days?

Minor and technical comments:

Page 2, line 30: The sentence starting with "Short-lived events of small number young Aitken particles" is difficult to understand, consider revising it. Does "small number" refer to low concentrations?

Page 3, line 4: should be "prior to reaching . . ."

Page 7, line 2: "non-refractive" should be "non-refractory"

Page 8, line 21: A more recent reference for NPF event classification is Kulmala et al. (2012).

Page 10, line 3: In the sentence " . . . have trace a lower number of . . ." the word "trace" should be omitted.

Page 10, line 21: As also suggested by the other referee, Section 3.5 could be divided into two parts, one dealing with CCN concentrations and another dealing with cloud droplet concentrations. That would make this section more readable.

Page 13, line 30: ". . . have a similar to ozone behavior . . ." should be ". . . behave similarly to ozone . . ."

References:

Kulmala, M., Petäjä, T., Nieminen, T., Sipilä, M., Manninen, H. E., Lehtipalo, K., Dal Maso, M., Aalto, P. P., Junninen, H., Paasonen, P., Riipinen, I., Lehtinen, K. E. J., Laaksonen, A., and Kerminen, V.-M.: Measurement of the nucleation of atmospheric aerosol particles. Nature Protocols 7, 1651–1667, 2012.

Su, L., Yuan, Z., Fung, J. C. H., Lau, A. K. H.: A comparison of HYSPLIT backward

trajectories generated from two GDAS datasets. Science of The Total Environment 506–507, 527–537, 2015. doi:10.1016/j.scitotenv.2014.11.072

---

## Author Comment (AC1) · 14 Oct 2016

We would like to thank both reviewers for their comments and recommendations. We believe that we have corrected and improved the paper by incorporating their comments, in the revised version. The figure proposed by the first reviewer was a very good idea where we had to clarify several points of 'our story' to provide sufficient context. We reran the simulations at higher resolution, replaced figures and modified the discussion, accordingly. The main changes are the following: Following both Reviewers' comment, regarding the model's estimation of the simulated new particle formation, we reran the model by ignoring NPF process. In the revised manuscript, section 3.5 is divided in 2 sections: 3.5 is called "Impact of NPF events on CCN production"

[Figure]

and 3.6 "Impact of NPF events on cloud droplet number." We followed first reviewerÎDs suggestion to use for the two types of northern flow the terms: Etesian Flow (EF) and Moderate Surface Flow (MSF), in order to have a more concise wording. We also followed the same formalism in the revised Tables and Figure captions.

Reviewer #1:

General Comments: Kalkavouras and co-authors present results from an intriguing experiment in the Aegean Sea. The nature of pollution arriving at the long-standing Finokalia measurement platform is investigated directly with observations at Santorini, a site strategically located along the trajectory from the European mainland. The observations target aerosol size distributions, relevant primary and secondary pollutant concentrations, and detection of new particle formation events. The authors identify two events representing characteristic flow from the north, albeit of two distinct types. The Etesian flow example is marked with NPF events at both sites, although the events are stronger at the Santorini site. The authors extend their observations of particle number and composition to predictions of CCN at various supersaturations. They also go further to predict the total effect of the NPF events on cloud droplet number, taking into account the impact of constrained available water vapor. The paper starts with a nice scientific idea and goes into good detail into the results. What I see lacking is a little more connective tissue linking the observations at each site with each other and the mainland into a cohesive story. The material is already there, but it is somewhat buried and could be highlighted with a figure, for instance. I would like to see the following points addressed before recommending publication:

Specific Comments: 1. I recommend the authors adopt two shorthand names for the distinct periods (22/7- 24/7 and 25/7-27/7). They could be referred to as "Etesian Flow" and "Moderate Surface Flow," for example. Small changes like this could help the readability of the paper. A useful addition to this paper would be a two-panel cartoon, each overlaid on the Greek domain map in Fig. 1, for example, that describes the factors at play in these two periods. They could identify generally where they expect

emissions, mixing, oxidation, NPF, and aging of new particles to be happening.

To be more concise, we used for the two types of northern flow the terms: Etesian Flow (EF) and Moderate Surface Flow (MSF). We also followed the same formalism in the revised Tables and Figure captions. Figure 1 is replaced with a two-panel cartoon in order to identify the locations where we expect/identify the various processes.

2. The WRF-Chem aerosol module configuration, as documented by the authors, is problematic for this particular application. It is quite likely that NPF events and subsequent processing are not captured realistically at all by the model. The sulfuric acid/water pathway parameterized by Kulmala et al. (1998) is likely not strong enough to enhance particles near the surface and lower troposphere to the levels observed at the Santorini site. It is now well-documented that other reagents play important roles in this process (e.g. NH3 and organics), and these pollutants have been identified by the authors to be present and significant components of the aerosol. My guess is that most of the Aitken-mode particles in the model originate from direct emissions, not from secondary generation. A related issue is the lack of a dedicated nucleation mode in the model. Without this mode, any NPF events will artificially broaden the Aitken mode distribution and give unrealistic lifetimes against deposition and coagulation. It will also affect the growth rates predicted by the model. The authors astutely sidestep relying on the model to predict size distributions and use their own observations when possible for calculating CCN and cloud drop number concentrations. However, since they include an entire section (2.2) detailing the regional modeling they performed, it is a good idea to explicitly state the limitations of this analysis for particle size distributions, and remind the reader that they are using the WRF-Chem output for its knowledge of advection flows and chemical composition, not microphysical processing.

We agree with the Reviewer's comment. In version 3.3 the aerosol models are not appropriate to simulate the NPF events realistically. Luo and Yu (2011), discuss the need to improve the representation of the nucleation process in earlier versions of WRF-Chem. We believe that more research is needed regarding the nucleation modeling

in the area, which we plan to perform in a separate paper in the near future. Nevertheless, we conducted another simulation ignoring the nucleation parameterization in order to comprehend/emphasize the spatial extent of these processes and present them in Fig. 1. In the revised manuscript, the relevant discussion on model limitations (due to lack of a dedicated nucleation mode, nucleation parameterization) is presented in section 2.3 'Regional modeling' (page 7, lines 6-21).

To further elucidate the conditions under which NPF events take place in our region, we reran the model by ignoring the NPF process. In the revised manuscript our hypothesis is not based on the simulated Aitken-mode particles but on the number concentration differences considering and ignoring NPF process. The relevant discussion is presented in section 3.4 'Spatial extent of NPF event (pages 13-16).

3. The authors conclude that the NPF events observed at Santorini are regional in nature with a spatial scale of 250 km and characteristic transport time of 4.5 hours. They also assert that Finokalia does not see the bursts because it is 3 hours away and particle populations age before they arrive there. The authors do note that this second observation demonstrates how site-to-site variability can be important during a regional event. I am not sure that this totally addresses the issue though. Why are the events sort of regional and sort of not-regional? Is this an issue of using up the NPF precursors before the air mass gets to Santorini and then shifting to chemical conditions that favor condensation to available surface area? Or is something else at play here?

We used the regional characterization, mainly because the number of particles remained high for several hours at Santorini (Kulmala et al., 2012). In addition, the NPF event was found to extend over hundreds of kilometers. Thus, as the reviewer points out, we tried to relate these fine aerosols to regional sources of pollution transferred by long-range transport (LRT) during Etesian flow conditions. Despite that, we observed local variability at sub-regional scales, due to the differences in geographical and atmospheric conditions between stations along the same trajectory. This is the

case between Santorini and Finokalia stations. We also expect that local variability is unavoidable at smaller scales, over locations at the same distance from the sources. This is based on the simulations that show that the spatial differences of chemical and physical properties in the initial steps of the formation, under the stable Etesian flow, produce streams with different characteristics, especially upwind of Santorini.

4. The paragraph beginning line 13 on page 9 describes an interesting hypothesis for how pollutants are transported to the middle of the Aegean Sea with limited aging. However, I'm not convinced there is enough evidence to warrant the detailed discussion that is given to this possibility or the certainty with which it is treated in the conclusions section. As described in my first comment, any model data related to the size distribution of Aitken-mode particles probably cannot be trusted in this case. If I understand correctly, the main assertion here is that the particles were formed over the Turkish mainland and transported quickly before they have a chance to be significantly coagulated away. Why could the enhanced number concentrations not come instead from oxidation and NPF over the water during transport, where there may be enhanced photochemistry, complex interactions with clouds, interesting boundary layer phenomena, etc? If I'm not understanding the meat of the argument correctly, please explain it to me and consider rephrasing it in the text to be clearer. What insight do the model CO concentrations help to provide regarding the stratification and mixing of distinct layers downwind of the continent?

The revised section 3.4 'Spatial extent of NPF event' (pages 13-16) explains more clearly now the processes taking place. The discussion is mainly based on the number concentration differences considering and ignoring NPF process and not on the Aitken-mode simulated particles. Although the model severely underestimates the NPF, the decisive role of the Etesian flow on the evolution of the phenomenon over the Aegean Sea is evident (from page 13 line 24 to page 15 line 1). The atmospheric conditions under a similar Etesian event have been studied thoroughly in a separate paper that has been submitted to BLM. In particular, the heat fluxes simulated and calculated

from airborne measurements over the AS (Tombrou et al., 2015) varied from -25 W m-2 (over the northeastern AS) to 25 W m-2 (over the southeastern AS). Furthermore, vertical cross-sections of measured CO concentrations along the eastern AS under an Etesian flow, are shown in Fig. 7 by Tombrou et al. (2015). The strong gradient of stratification and mixing downwind of AS, is apparent. In particular, at $40°$ latitude, where the plume leaves the Turkey continent, the vertical mixing extends up to 500-600m height according to the CO vertical extent. Above the Cyclades complex (lat $36.5°$ - $38°$) the mixing extends up to 1km and gradually increases up to 2km, upwind of Crete (Finokalia at $35°$).

5. I recommend separating the paragraphs detailing the Nd calculations (starting on Page 12) into their own section, perhaps called "Impact of NPF events on cloud droplet number." Then section 3.5 would be called "Impact of NPF events on CCN production."

Done

6. How is the partial sensitivity of cloud droplet number to chemical composition and vertical velocity determined? Can the equations be provided? What is the uncertainty associated with this? Please document it if possible.

The reviewer raises a good question. The sensitivity is derived from the parameterization using either a direct sensitivity or finite difference approach, as described in Karydis et al. (2012). Here we use the finite difference implementation. This information is now given on page 6; Ln 16-18. The accuracy of the method, i.e. the ability of the parameterization to capture the sensitivity of droplet number to each parameter examined was explored in detail by Morales and Nenes (2014). Given that the parameterization has been shown to give cloud droplet closure in ambient clouds to within experimental uncertainty (Meskhidze et al., 2005; Fountoukis et al., 2007; Hoyle et al., 2016), and that the same parameterization also reproduces the droplet number and sensitivities of the detailed numerical simulation with high fidelity (Morales and Nenes, 2014), we expect the sensitivities and attribution calculations presented here to be

representative of the ambient clouds in the study region.

Minor Changes/Typos: Pg 2, Ln 25: The phrase "without any particular seasonal prefer­ence about their occurrence" is difficult to understand. Can the authors please reword this to be more specific?

The phrase has been replaced by (Pg 3, Ln 11-12): "Most of these ground-based observations indicate that the mass of fine aerosols presents a summer maximum, however the frequency of the events is season independent."

Pg 3, Ln 4: "prior to reaching"

Done

Pg 5, Ln 11-13: This is technically not a sentence.

Replaced by (page 8, lines 6-8): "Air mass origin and trajectories were de­termined by HYSPLIT4 (Hybrid Single-Particle Lagrangian Integrated Trajectory; www.arl.noaa.gov/ready/ hysplit4.html) back-trajectory analysis (Draxler and Rolph, 2015)."

Fig. 3 and caption: "open circles" not "cycles".

Done.

Also, please indicate on the figure that the solid lines describe wind speed and the circles describe direction. It is hinted at in the figure and explained in the caption, but it would be quicker for the reader to have it identified visually, with an arrow or something.

Figure 3 is replaced by a new one, where the abbreviations 'ws' and 'wd' are now in­cluded indicating wind speed and wind direction respectively. The caption is rephrased accordingly: 'Time series of the wind speeds (ws, solid lines on left axis) and wind directions (wd, open circles on right axis) at Santorini (simulations by the WRF-Chem model) and at Finokalia (measurements). The second period of the EF is shaded with yellow and the MSF period with grey.

Pg 6, Ln 16-17: The "less pronounced" diurnal cycle at Finokalia for ozone is not obvious to me from Fig. S4. Please include a plot of the actual diurnally averaged profiles or report the daily minima and maxima to demonstrate this point.

We refer to the Etesian period (EF) that corresponds to the yellow panel in Fig. S4. The mean diurnal range at Finokalia station (from 21 to 24 July) is 8 ppbv, while at Santorini, for the same period is 18 ppbv (Fig. S4). This information is now included in the text (page 9, lines 22-24)

Pg 6, Ln 28-29: I would not characterize -21% or -15% under-prediction as "small". Either establish what they are small compared to, or please get rid of this qualification.

We agree with the reviewer's comment, therefore, we decided to delete the word 'small'.

Pg 6, Ln27-30: Please break this sentence up. It is long and confusing.

Replaced by (page 10, lines 9-13): Simulations confirm that the air masses received at both stations during the prevailing strong northern wind are of the same origin, and representative of EF conditions (Fig. S3) albeit with an O3 under-prediction (average bias during afternoon hours up to -21% on 23 and -15% on 24 July, Fig. 5). During the MSF period, simulations indicate an O3 increase, especially in the southern AS, but also underpredicted (average bias during afternoon hours up to -24% on 26 July, Fig. 5).

Pg 6, Ln 31-32: Is there a more recent or relevant reference than McKeen et al. (1991)? Anything that specifically identifies this model scenario or modern European scenarios in general as suffering from ozone boundary condition issues?

The chemical boundary conditions used in this modeling study are hardcoded in the WRF-Chem model. The values are based on an idealized, northern hemispheric, mid-latitude, clean environmental, vertical profile from the NOAA Aeronomy Lab Regional Oxidant Model (NALROM) (Liu et al. 1996; Peckham et al. 2011). This information has been added to section 2.3 'Regional modeling' (page 7, lines 25-27 while the reference

of McKeen et al. (1991) was omitted.

Pg 7, Ln 5-6: In what way did the inorganic and organic mass concentrations show "similar behavior" to that of ozone? Are the authors just identifying them all as secondary pollutants? Please provide an estimate of the correlation coefficient or index of agreement for a statement like this.

We identify all of them as secondary pollutants driven by the same meteorological conditions. The R2 of O3 to Organics and O3 to inorganics is 0.5 and 0.59 respectively (included in the revised manuscript on page 10, line 22). Also, the simulated spatial patterns of O3 and sulfates are similar, for each period (EF and MSF).

Pg 7, Ln 9: Please remove the comma after the parentheses.

Done

Pg 7, Ln 20: Please refer to some quantitative statistics to back up this claim.

An extended evaluation of WRF-Chem model against airborne and ground observations over the AS during the Etesians is presented in Bossioli et al. (2016). In that study biomass burning emissions were also included. After the reviewer's suggestion some statistics have been added in the revised manuscript: For EF period (Page 11, lines 3-4): "….on average during EF underprediction of 30% for sulfates and 60% for ammonium" For MSF period (Page 11, lines 12-15): "(simulated and observed concentrations correlate during both periods R2=0.8), however they are lower than the measured values at Finokalia station (on average underprediction of 50% for sulfates and 75% for ammonium).

Pg 10, Ln 6-8: This sentence is worded in a confusing way.

The discussion has been revised (page 15, lines 10-17). The specific sentence has been revised to "The nucleation-mode particles are significantly reduced as they have shifted gradually towards larger sizes (Aitken-mode), before reaching Finokalia (Fig. 4)." Pg 10, Ln 25 – Pg 11, Ln 19: Most of this material would be better-placed in the

methods section (2.3 maybe). This goes for the second paragraph a page 12 as well.

Done

References:

Bossioli, E., Tombrou, M., Kalogiros, J., Allan, J., Bacak, A., Bezantakos, S., Biskos, G., Coe, H., Jones, B. T., Kouvarakis, G., Mihalopoulos, N., and Percival, C. J.: Atmospheric composition in the Eastern Mediterranean: Influence of biomass burning during summertime using the WRF-Chem model, Atmos. Environ., 132, 317–331, 2016

Draxler, R.R. and Rolph, G. D.: HYSPLIT (HYbrid Single-Particle Lagrangian Integrated Trajectory) Model access via NOAA ARL READY Website (http://ready.arl.noaa.gov/HYSPLIT.php). NOAA Air Resources Laboratory, Silver Spring, MD., 2015

Fountoukis, C., Nenes, A., Meskhidze, N., Bahreini, R., Brechtel, F., Conant, W. C., Jonsson, H., Murphy, S., Sorooshian, A., Varutbangkul, V., R. C. Flagan, and J. H. Seinfeld Aerosol–cloud drop concentration closure for clouds sampled during ICARTT, J.Geoph.Res., 112, D10S30, doi:10.1029/2006JD007272, 2007 Hoyle, C.R., Webster, C.S., Rieder, H.E., Nenes, A., Hammer, E., Herrmann, E., Gysel, M., Bukowiecki, N., Weingartner, E., Steinbacher, M., and U. Baltensperger Chemical and physical influences on aerosol activation in liquid clouds: a study based on observations from the Jungfraujoch, Switzerland, Atmos.Chem.Phys., 16, 4043–4061, 2016 Karydis, V. A., Capps, S. L., Russell, A. G., and Nenes, A.: Adjoint sensitivity of global cloud droplet number to aerosol and dynamical parameters, Atmos. Chem. Phys., 12, 9041-9055, doi:10.5194/acp-12-9041-2012, 2012 Kulmala, M., Petäjä, T., Nieminen, T., Sipilä, M., Manninen, H. E., Lehtipalo, K., Dal Maso, M., Aalto, P. P., Junninen, H., Paasonen, P., Riipinen, I., Lehtinen, K. E. J., Laaksonen, A., and Kerminen, V.-M.: Measurement of the nucleation of atmospheric aerosol particles. Nature Protocols 7, 1651–1667, 2012.

Liu, S. C., McKeen, S. A., Hsie, ÎŢ-Îě., Lin, ., Kelly, K. K., Bradshaw, J. D., Sandholm, S. T., Browell, E. V., Gregory, G. L., Sachse, G. W., Bandy, A. R., Thornton, D. C., Blake, D. R., Rowland, F. S., Newell, R., Heikes, B. G., Singh, H., Talbot, R. W.: Model study of tropospheric trace species distributions during PEM-West A, J. Geophys. Res., 101(D1), 2073–2085, doi:10.1029/95JD02277, 1996 Luo, G. and Yu, F.: Simulation of particle formation and number concentration over the Eastern United States with the WRF-Chem + APM model, Atmos. Chem. Phys., 11, 11521-11533, doi:10.5194/acp-11-11521-2011, 2011. Meskhidze, N., A. Nenes, Conant, W. C., and Seinfeld, J.H.: Evaluation of a new Cloud Droplet Activation Parameterization with In Situ Data from CRYSTAL-FACE and CSTRIPE, J.Geoph.Res., 110, D16202, doi:10.1029/2004JD005703, 2005 Morales Betancourt, R., and Nenes, A.: Aerosol Activation Parameterization: The population splitting concept revisited, Geosci.Mod.Dev., 7, 2345–2357, 2014 Peckham, S., G. A. Grell, S. A. McKeen, M. Barth, G. Pfister, C. Wiedinmyer, J. D. Fast, W. I. Gustafson, R. Zaveri, R. C. Easter, J. Barnard, E. Chapman, M. Hewson, R. Schmitz, M. Salzmann, S. Freitas: WRF/Chem Version 3.3 User's Guide. NOAA Technical Memo., 98 pp., 2011

Tombrou, M., Bossioli, E., Kalogiros, J., Allan, J. D., Bacak, A., Biskos, G., Coe, H., Dandou, A., Kouvarakis, G., Mihalopoulos, N., Percival, C. J., Protonotariou, A. P., and Szabó-Takács, B.: Physical and chemical processes of air masses in the Aegean Sea during Etesians: Aegean-GAME airborne campaign, Sci. Total Environ., 506-507, 201–216, doi:10.1016/j.scitotenv.2014.10.098, 2015

---

## Author Comment (AC2) · 14 Oct 2016

We would like to thank both reviewers for their comments and recommendations. We believe that we have corrected and improved the paper by incorporating their comments, in the revised version. The figure proposed by the first reviewer was a very good idea where we had to clarify several points of 'our story' to provide sufficient context. We reran the simulations at higher resolution, replaced figures and modified the discussion, accordingly. The main changes are the following: Following both Reviewers' comment, regarding the model's estimation of the simulated new particle formation, we reran the model by ignoring NPF process. In the revised manuscript, section 3.5 is divided in 2 sections: 3.5 is called "Impact of NPF events on CCN production" and 3.6

"Impact of NPF events on cloud droplet number." We followed first reviewer suggestion to use for the two types of northern flow the terms: Etesian Flow (EF) and Moderate Surface Flow (MSF), in order to have a more concise wording. We also followed the same formalism in the revised Tables and Figure captions.

Reviewer#2 The manuscript presents measurements of the number size distribution and chemical composition of submicron aerosols at two islands in the Eastern Mediterranean. The analysis is based on a measurement period over two weeks in the summer 2013, during persistent transport of continental airmasses from north to the sites. A chemical transport model and airmass back-trajectories are used to identify the source areas and transport routes of aerosols to the sites. Using case studies of two new particle formation (NPF) events the contribution of NPF to both the cloud condensation nuclei (CCN) and cloud droplet (Nd) concentrations is assessed. The results for CCN and Nd are based on Köhler theory and parameterizations. I agree with the comments presented by the anonymous referee #1, and would like the authors to address my further comments below. After addressing these comments I can recommend the manuscript for publication in Atmospheric Chemistry and Physics.

General comments: Page 5, lines 6–7: Is it known what are the possible reasons for the underestimation of organic matter concentrations in the model results; could it be due to underestimation of primary emissions or underestimation of SOA formation in the model? The biases are probably related to the underestimated POA emissions but also to the limitation of the RADM2 mechanism regarding the treatment of monoterpene emissions (Tuccella et al., 2012). This information is now included on page 8, lines 4-5. WRF-Chem simulations over the Aegean Sea during Etesian flow revealed that the simulated SOA, formed from anthropogenic and biogenic emissions, contributes respectively to less than 5% and almost negligibly to the OM (Bossioli et al., 2016). The importance of secondary aerosols over the area has been pointed out in earlier works (Athanasopoulou et al., 2015; Fountoukis et al., 2011)

Page 5, lines 8–12: Care should be taken when using the HYSPLIT model with the

GDAS 0.5_ input data: the back-trajectory results might differ from those obtained with GDAS 1_ input data due to the differences in the airmass vertical advection calculation method between these two datasets (see e.g. Su et al., 2015). Perhaps the authors could check that their back-trajectories shown in Figure 2 remain the same if using the GDAS with 1_ resolution as input meteorological data.

There are no significant differences, especially at low levels. The differences are mainly noticed on the 24th, but they do not change the hypothesis that air masses are better mixed throughout the boundary layer, covering a broader area over Asian Turkey. See attached figure.

Page 8, lines 6–7: Why are coagulation losses not included in the calculation of the formation rate of nucleation mode particles? This should be fairly straightforward to calculate based on the measured size distributions, and including the coagulation losses would make the calculated formation rates more readily comparable to literature values (which typically account for coagulation).

Both coagulation flux and condensational growth are now included in the calculations. The text has been modified accordingly (page 12, lines 10-14).

Page 10, line 4: Where does the 3 hour difference in the comparison between particle observations at Santorini and Finokalia come from? Based on the particle size distribution data in Fig. 8 the particle formation at both stations seems to start at 9 a.m. on 23 July, and the only appreacable difference in the particle concentrations in Fig. 4 seems to be in the nucleation mode concentration (i.e. intensity of particle formation). Regarding the discussion on the CCN-sized particles and the calculated hygroscopicity parameters, it would be interesting to see how the results differ on days without new particle formation. This type of comparison between NPF and non-NPF days would put the results presented in the manuscript better into context with regard to the importance of NPF to CCN and cloud droplet number at the Aegean Sea. Where there during the campaign any such non-NPF days for which the parameters of Table 3 could

be calculated and reported for comparison with the two NPF days?

We agree with the reviewer that this was not clear in the text. The air masses spent 3-4 h to reach Finokalia after Santorini, according to HYSPLIT (Fig. S3 left panel), on 23 July. The 3-h transit timescale is in agreement with the prevailing wind speed (about 10 m s-1; Fig. S1) and the 120 km distance between Santorini and Finokalia. For this reason, we claim that the air masses reaching Finokalia earlier (Fig. 4) are probably due to a local nucleation event initiated at Heraklion (Crete).

Throughout the non-NPF events (MSF period), the CCN concentrations decrease by almost 48% and 23% at Santorini and Finokalia respectively, compared to the levels during the NPF events. We have added this information on page 17 (lines 22-24) but we decided not to change Table 3.

Minor and technical comments:

Page 2, line 30: The sentence starting with "Short-lived events of small number young Aitken particles" is difficult to understand, consider revising it. Does "small number" refer to low concentrations?

This sentence has been replaced by (page 3, lines 18-19): "A few short-lived particle formation events (18–25 nm) were first recorded at Finokalia by Kalivitis et al. (2008), arriving with low speed from the west, during autumn."

Page 3, line 4: should be "prior to reaching "

Done

Page 7, line 2: "non-refractive" should be "non-refractory"

Done

Page 8, line 21: A more recent reference for NPF event classification is Kulmala et al. (2012).

Done

Page 10, line 3: In the sentence " : : : have trace a lower number of : : :" the word "trace" should be omitted.

Done

Page 10, line 21: As also suggested by the other referee, Section 3.5 could be divided into two parts, one dealing with CCN concentrations and another dealing with cloud droplet concentrations. That would make this section more readable.

Done

Page 13, line 30: ": : : have a similar to ozone behavior : : :" should be ": : : behave similarly to ozone : : :"

Done

References:

Athanasopoulou E., E.P. Protonotariou, E. Bossioli, A. Dandou, M. Tombrou, J.D. Allan, H. Coel, N. Mihalopoulos, J. Kalogiros, A. Bacak, J. Sciare, G. Biskos: 'Aerosol chemistry above an extended Archipelago of the Eastern Mediterranean basin during strong northern winds', Atmospheric Chemistry and Physics, 15, 8401-8421, doi: 10.5194/acp-15-8401-2015, 2015

Bossioli, E., Tombrou, M., Kalogiros, J., Allan, J., Bacak, A., Bezantakos, S., Biskos, G., Coe, H., Jones, B. T., Kouvarakis, G., Mihalopoulos, N., and Percival, C. J.: Atmospheric composition in the Eastern Mediterranean: Influence of biomass burning during summertime using the WRF-Chem model, Atmos. Environ., 132, 317–331, 2016

Fountoukis, C., P.N. Racherla, H.A.C. Denier van der Gon, P. Polymeneas, P.E. Haralabidis, A. Wiedensohler, C. Pilinis, and S.N. Pandis, 2011: Evaluation of a three-dimensional chemical transport model (PMCAMx) in the European domain during the EUCAARI May 2008 campaign. Atmos. Chem. Phys., 11, 10331-10347,

doi:10.5194/acp-11-10331-2011, 2011

Kalivitis, N., Birmili, W., Stock, M., Wehner, B., Massling, A., Wiedensohler, A., Gerasopoulos, E. and Mihalopoulos, N.: Particle size distributions in the Eastern Mediterranean troposphere, Atmos. Chem. Phys., 8, 6729–6738, doi:10.5194/acp-8-6729-2008, 2008

Please also note the supplement to this comment:
http://www.atmos-chem-phys-discuss.net/acp-2016-330/acp-2016-330-AC2-supplement.pdf

[Figure]

**Supplement:**

---

## Referee Report (RR1)

Review of the revised submission of the manuscript

"New particle formation in the South Aegean Sea during the Etesians: importance for CCN production and cloud droplet number"

by Kalkavouras et al., submitted to Atmospheric Chemistry and Physics (acp-2016-330)

In the revised version of the manuscript, the authors have addressed all the comments and suggestions for improvement from both reviewers. The manuscript includes now a more clear discussion on the limitations in the aerosol dynamics of the WRF-Chem regional model, especially related to the modelling of new particle formation (NPF) which is one of the focus points of the manuscript. The comparison between NPF and non-NPF days has been made more extensive regarding both the regional modelling results and the measured aerosol parameters (aerosol number and mass concentrations and CCN-proxy concentrations).

I recommend the revised manuscript to be published in Atmospheric Chemistry and Physics.